# Fast and Highly Expressive Policy Learning for Offline Reinforcement Learning via Bootstrapped Flow Q-Learning

**Thanh Nguyen** [1]  **Tri Ton** [1]  **Hongbin Choe** [1]  **Tung M. Luu** [1]  **Chang D. Yoo** [1]

## Abstract

Diffusion-based Q-learning has emerged as a powerful paradigm for offline reinforcement learning, but its reliance on multi-step denoising makes both training and inference computationally expensive and brittle. Recent efforts to accelerate diffusion Q-learning toward single-step action generation typically introduce auxiliary networks, policy distillation, or multi-phase training, which frequently compromise simplicity, stability, or performance. To address these limitations, we introduce Bootstrapped Flow Q-Learning (BFQ), a novel framework that enables accurate single-step action generation during both training and inference—without auxiliary networks or distillation procedures. BFQ adopts a divide-and-conquer view of the displacement vector along the flow path: it begins by learning short-range displacements that can be accurately estimated from the Flow Matching marginal velocity, and bootstraps these components to directly learn a noise-to-action mapping in a single step. This formulation eliminates multi-step denoising, resulting in a learning procedure that is substantially faster, simpler, and more robust. Extensive D4RL evaluations show that BFQ improves performance while significantly reducing computational cost compared to multi-step diffusion baselines, demonstrating that single-step action generation suffices for high-performance offline Reinforcement Learning.

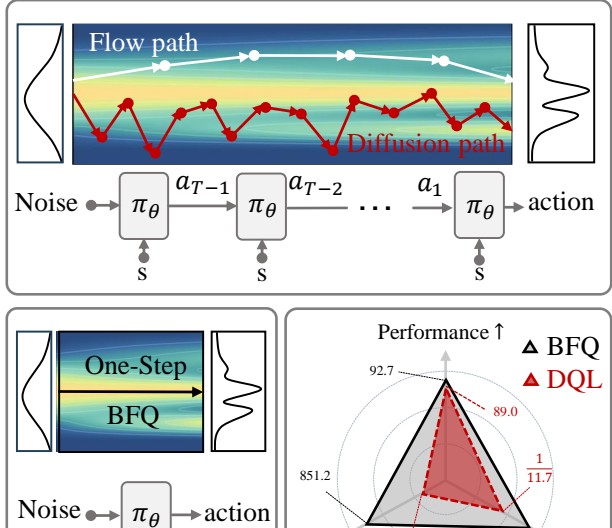

*Figure 1.* Diffusion/Flow Policy typically relies on multi-step denoising, requiring multiple forward evaluations during action generation and necessitating policy optimization via backpropagation through time (BPTT), which significantly slows down training, deployment and leads to more fragile optimization. In contrast, BFQ learns a direct noise-to-action mapping that preserves action expressiveness, enabling faster inference and training with direct, BPTT-free gradients. The radar chart indicates average results on D4RL locomotion tasks, demonstrating that BFQ achieves higher action frequency (Hz), better training efficiency (measured as 1/training time), and even outperforms the well-known Diffusion Q-Learning.

## 1. Introduction

Offline reinforcement learning (Offline RL) enables the learning of decision-making policies from fixed datasets, allowing safe and scalable deployment in domains such as robotics, autonomous driving, and healthcare, where online exploration is costly or infeasible (Levine et al., 2020). One of the core challenges in offline RL is learning policies that are both expressive and computationally efficient: policies must capture complex, potentially multi-modal action distributions induced by diverse behaviors while remaining amenable to stable value-based optimization. Recent work has therefore turned to diffusion-based generative policies, which provide substantially greater expressiveness than conventional Gaussian policies (Fujimoto & Gu, 2021; Kumar

[1]Department of Electrical Engineering, KAIST, Daejeon, Korea. Correspondence to: Thanh Nguyen <thanhnguyen@kaist.ac.kr>, Chang D. Yoo <cd_yoo@kaist.ac.kr>.

*Proceedings of the 43rd International Conference on Machine Learning*, Seoul, South Korea. PMLR 306, 2026. Copyright 2026 by the author(s).

et al., 2020; Kostrikov et al., 2021) and have demonstrated strong empirical performance in offline RL (Dong et al., 2024). These advances open new directions for policy design by leveraging expressive generative modeling to capture diverse behaviors beyond simple unimodal assumptions (Dong et al., 2024; Wang et al., 2022).

Despite their strong empirical performance, diffusion-based policies typically rely on multi-step action generation and require backpropagation through time (BPTT) during training (Kang et al., 2023; Wang et al., 2022), as illustrated in Figure 1. At deployment, multi-step action generation reduces action frequency, limiting real-time responsiveness. During training, the combination of multi-step generation and BPTT not only increases computational cost but is also known to introduce optimization instability, which can hinder convergence to optimal performance (Park et al., 2025). Recent works have made partial progress toward mitigating these issues—for example, by designing more efficient denoising solvers (Kang et al., 2023), adopting IQL-based learning paradigms (Hansen-Estruch et al., 2023), or introducing auxiliary policies and policy distillation mechanisms (Park et al., 2025; Chen et al., 2023; 2024; Lu et al., 2025b), sometimes relying on Jacobian computations (Nguyen & Yoo, 2026). However, such methods typically incur additional algorithmic complexity, require multi-stage training pipelines, or introduce unfavorable trade-offs between scalability and policy quality.

These limitations primarily stem from the reliance on multi-step action generation in diffusion-based policies. A natural solution is to learn a single-step policy that preserves the expressiveness of generative models. Thus, we introduce Bootstrapped Flow Q-Learning (BFQ), a novel framework that enables highly expressive single-step action generation during both training and inference. BFQ eliminates the need for backpropagation through time (BPTT) while requiring neither auxiliary models, policy distillation, nor multi-stage training pipelines. More specifically, it is built upon the Flow Matching (Lipman et al., 2022), which provides a simpler and more efficient alternative to diffusion-based methods. However, standard Flow Matching learns marginal velocity fields, which often induce curved trajectories and thus limit the accuracy of single-step inference. BFQ resolves this limitation through a divide-and-conquer bootstrapping strategy that directly learns the flow endpoint mapping. Specifically, it exploits the fact that small displacements can be accurately estimated using marginal velocities from Flow Matching, and then bootstraps these short-range displacements to progressively learn a direct noise-to-action displacement in a single step. Moreover, BFQ employs a single shared policy network, trained from scratch, that jointly supports displacement bootstrapping and velocity anchoring to the underlying flow dynamics across all scales. This unified design removes the need for multi-step denoising and

BPTT, resulting in a learning procedure that is substantially faster, simpler, and more robust than prior diffusion-based approaches. Extensive experiments on the D4RL benchmark demonstrate that BFQ not only outperforms DQL in terms of return, but also achieves substantial improvements in both training and inference efficiency while maintaining a streamlined learning pipeline. Overall, BFQ delivers consistently strong performance, establishing it as a fast and state-of-the-art method on D4RL.

## 2. Related Work

**Diffusion Models in Offline Reinforcement Learning (Offline RL).** Offline RL seeks to derive effective decision-making policies solely from pre-collected datasets, without requiring additional interaction with the environment (Levine et al., 2020). Early offline RL algorithms mitigate distributional shift by introducing conservative learning objectives, as exemplified by CQL (Kumar et al., 2020), TD3+BC (Fujimoto & Gu, 2021), and IQL (Kostrikov et al., 2021). While effective, these methods typically employ unimodal Gaussian policies, which are inherently limited in their ability to capture the complex, multimodal action distributions arising from diverse behavior data. To overcome these representational constraints, recent work has incorporated diffusion models (Ho et al., 2020; Song et al., 2020) into offline RL. Owing to their strong capacity for modeling high-dimensional and multimodal distributions, diffusion models have been explored in multiple roles: as planners for trajectory generation (Janner et al., 2022; Ajay et al., 2022), as expressive policy parameterizations (Wang et al., 2022; Hansen-Estruch et al., 2023), and as generative data augmentation mechanisms (Zhu et al., 2023). Among these approaches, Diffusion Q-Learning (DQL) (Wang et al., 2022) has emerged as a particularly competitive baseline by replacing the Gaussian policy in TD3+BC with a diffusion-based policy to better represent multimodal action spaces. Subsequent evaluations, including Clean Diffuser (Dong et al., 2024) and recent large-scale empirical studies (Lu et al., 2025a;b), consistently demonstrate DQL's superiority over both policy-based and planner-based alternatives. Nevertheless, DQL incurs substantial computational cost during training and inference (Kang et al., 2023) and can suffer from unstable or suboptimal convergence behavior (Park et al., 2025).

A number of subsequent works have sought to alleviate these shortcomings. Early efforts reduce computational cost by employing more efficient solvers to decrease the number of denoising steps required by diffusion policies (Kang et al., 2023). Other approaches avoid backpropagation through time (BPTT) in DQL by training a diffusion policy to imitate the behavior policy, while reweighting actions using a separately learned IQL-based value func-

tion (Hansen-Estruch et al., 2023). Further extensions apply policy distillation to obtain a single-step policy from a multi-step diffusion model (Chen et al., 2023; 2024). However, methods built upon IQL-style learning are generally less effective than actor–critic approaches (Park et al., 2024). To bridge this gap, (Park et al., 2025) adopts a flow-based model to clone the behavior policy and subsequently distills it into a one-step policy suitable for actor–critic updates. Although this approach enables single-step inference, the distillation process still requires repeated queries to the underlying multi-step diffusion or flow model. In addition, other approaches rely on average-velocity estimation based on MeanFlow (Nguyen & Yoo, 2026; Geng et al., 2025), which entails explicit Jacobian computations.

Overall, while these methods partially mitigate the inefficiency of DQL, they introduce auxiliary models, additional optimization stages, or multi-phase training pipelines, substantially increasing system complexity and limiting practical generality. In contrast to prior work, we identify the diffusion policy itself as the principal source of inefficiency and instability in DQL. We address this issue directly by learning a highly efficient single-step policy that retains expressiveness while eliminating auxiliary policies, distillation procedures, and Jacobian computations, resulting in a simpler but remain robust solution.

## 3. Preliminaries

**Offline Reinforcement Learning.** Reinforcement learning (RL) is commonly formulated as a Markov Decision Process (MDP) $\mathcal{M} = (\mathcal{S}, \mathcal{A}, P, R, \gamma)$, where $\mathcal{S}$ and $\mathcal{A}$ denote the state and action spaces, $P(s' \mid s, a)$ specifies the transition dynamics, $R(s, a)$ is the reward function, and $\gamma \in [0, 1)$ is the discount factor that balances immediate and future rewards. The objective of RL is to learn a parameterized policy $\pi_\theta(a \mid s)$ that maximizes the expected discounted return:

$$\mathbb{E}_\pi \left[ \sum_{h=0}^{\infty} \gamma^h R(s_h, a_h) \right]. \qquad (1)$$

To support policy optimization, the action–value function under policy $\pi$ is defined as:

$$Q^\pi(s, a) = \mathbb{E}_\pi \left[ \sum_{h=0}^{\infty} \gamma^h R(s_h, a_h) \,\middle|\, s_0 = s, \ a_0 = a \right], \quad (2)$$

which represents the expected cumulative return obtained by taking action $a$ in state $s$ and subsequently following policy $\pi$.

Offline RL considers the setting where the agent cannot interact with the environment and must instead learn from a fixed dataset of transitions $\mathcal{D} = \{(s_h, a_h, s_{h+1}, r_h)\}$. The central challenge is to learn an effective policy solely from this static dataset, which often contains suboptimal or heterogeneous behaviors, without the ability to perform additional exploration.

**Behavior-regularized actor-critic.** Behavior-regularized actor–critic methods (Wu et al., 2019; Fujimoto & Gu, 2021) constitute one of the simplest yet most effective frameworks for offline RL, and are widely adopted in diffusion-based policy learning. The framework alternates between optimizing a critic (*i.e.,* Q function) that estimates action values (*i.e.,* policy evaluation) and optimizing an actor (*i.e.,* $\pi_\theta$) that learns a policy constrained to remain close to the data distribution (*i.e.,* policy improvement).

Specifically, the critic is trained by minimizing the critic loss:

$$\mathcal{L}_Q(\phi) = \mathbb{E}_{\mathcal{D}, a' \sim \pi_{\theta'}} \left[ \left( r + \gamma \min_i Q_{\phi'_i}(s', a') - Q_{\phi_i}(s, a) \right)^2 \right],$$
$$(3)$$

where $i \in 1, 2$ indexes the two Q networks for double Q-learning, and $(\phi', \theta')$ denote target network parameters updated via exponential moving average (EMA) (Fujimoto & Gu, 2021).

The actor is optimized to maximize the estimated value while remaining close to the behavior policy with the actor loss:

$$\mathcal{L}(\theta) = -\mathbb{E}_{(s,a) \sim \mathcal{D}, a^\pi \sim \pi_\theta} \left[ \alpha Q_\phi(s, a^\pi) + \log \pi_\theta(a|s) \right],$$
$$(4)$$

where $\alpha$ controls the strength of value maximization. The first term encourages high-value actions, while the log-likelihood term acts as a behavior cloning regularizer that prevents the learned policy from deviating excessively from the dataset distribution.

Importantly, during the actor update, gradients from the actor loss are propagated to the policy parameters through the fixed Q-function. Consequently, because diffusion-based policies rely on multiple recursive denoising steps to generate a clean action, computing gradients from the final action back to the model parameters necessitates backpropagation through time (BPTT).

Furthermore, both the actor and critic objectives require sampling actions from the diffusion policy, which involves multi-step generation. This significantly slows down the computation of both objectives during training.

**Modeling Policy with Flow Matching.** Diffusion models generally generate samples by simulating stochastic differential equations (SDEs), which typically induce stochastic and curved trajectories along the denoising path. This property makes extending diffusion models to reliable single-step sampling inherently challenging. Flow Matching (FM) (Lipman et al., 2022) offers a principled alternative by learning deterministic ordinary differential equations (ODEs) that

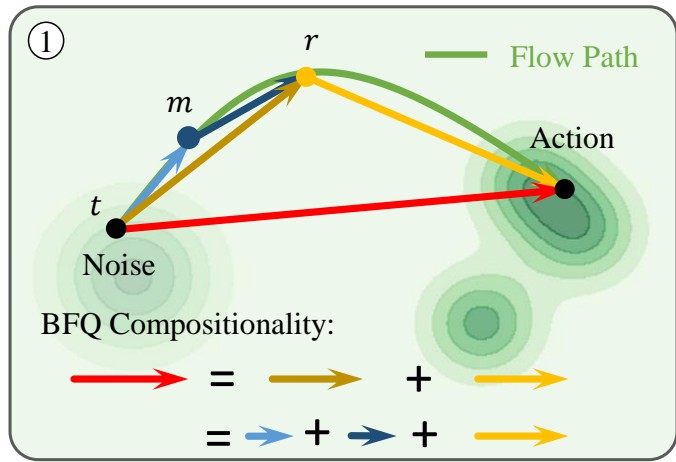
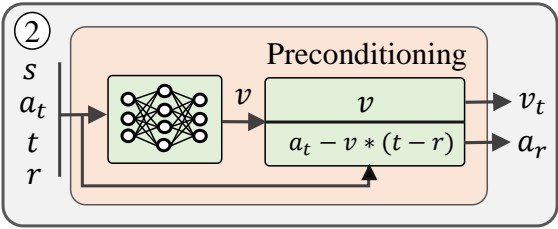
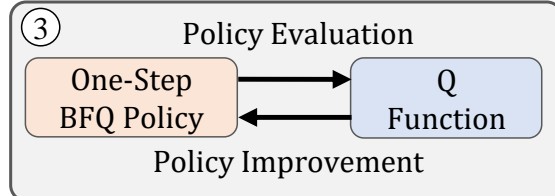

*Figure 2.* Illustration of the BFQ framework. (1) Standard flow-based policies learn marginal velocity fields, which often induce curved generation trajectories, illustrated by the green vectors. In contrast, BFQ directly learns a flow endpoint map, shown by the red arrow, via a divide-and-conquer bootstrapping strategy. BFQ leverages the fact that small displacements can be accurately estimated from marginal velocities in flow matching, and progressively bootstraps these short-range displacements (e.g., combining two blue vectors to form a yellow vector, and so on) to learn a single-step noise-to-action displacement. (2) Overview of the policy architecture. BFQ employs a simple preconditioning mechanism that anchors the policy to marginal velocities, automatically satisfies the identity condition, and remains mathematically well-founded and flexible for policy learning. (3) BFQ is built on top of a behavior-regularized actor–critic framework with single-step action generation, enabling fast inference while eliminating backpropagation through time required by diffusion-based policies.

map noise directly to data along smoother and more direct trajectories (Liu et al., 2022). Consequently, modeling policies via FM has the potential to improve sampling efficiency and provides a more amenable framework for constructing single-step policies.

A common policy parameterization follows a flow-matching formulation with linear interpolation paths and uniform time sampling. Given a dataset $(s, a) \sim \mathcal{D}$ and noise $\epsilon \sim \mathcal{N}(\mathbf{0}, \boldsymbol{I})$, the linear flow path $a_t$ and its corresponding conditional velocity $v_{cond}(a_t \mid a, \epsilon; s)$ are defined as:

$$a_t = (1-t)a + t\epsilon, \qquad v_{cond} = \epsilon - a, \qquad t \in [0, 1]. \quad (5)$$

By construction, $a \equiv a_0$, and we omit the subscript when referring to the clean action for simplicity.

FM essentially learns *the marginal velocity field*, parameterized by a neural network $v_\theta(a_t, t; s)$, by minimizing the Conditional Flow Matching (CFM) objective:

$$\mathcal{L}_{\text{CFM}}(\theta) = \mathbb{E}_{t \sim U[0,1], \mathcal{D}, \epsilon} \left\| v_\theta(a_t, t; s) - v_{cond}(a_t \mid a, \epsilon; s) \right\|^2. \quad (6)$$

At inference time, an action for state $s$ is generated by solving the ODE:

$$\frac{da_t}{dt} = v(a_t, t; s), \quad \text{initialized from } a_1 \equiv \epsilon \sim \mathcal{N}(\mathbf{0}, \boldsymbol{I}), \quad (7)$$

which is typically approximated using numerical solvers such as Euler's method:

$$a_{t-\Delta} = a_t - \Delta * v(a_t, t; s), \quad (8)$$

where $\Delta$ is the Euler step size.

Intuitively, one might expect that a flow-based policy would enable reliable single-step action generation by setting $\Delta t = 1$. However, in practice, the learned marginal velocity field often induces curved global trajectories, as illustrated by the green arrow in Figure 2 (1), which makes single-step generation inaccurate. Consequently, although flow-based policies are more efficient than diffusion-based counterparts, they still require multiple integration steps to produce reliable actions. As a result, when integrated into the actor–critic framework, policy optimization continues to depend on backpropagation through time (BPTT) and additional mechanisms are needed when attempting single-step action generation.

In this work, we further extend flow-based policies and develop a variant of the behavior-regularized actor–critic framework for offline RL that enables single-step action generation for both policy improvement and policy evaluation.

## 4. Methodology

We now introduce Bootstrapped Flow Q-learning (BFQ). Our design is guided by three desiderata: (i) has a similar expressiveness compared to the diffusion policy to model complex, multimodal action distributions; (ii) enable single-step action generation for eliminating backpropagation through time (BPTT) and accelerating training and inference; (iii)

retain a simple and practical formulation, using a single policy network without distillation, multi-stage training, or Jacobian calculation.

Our key idea departs from standard flow-matching formulations that model marginal velocity fields. Instead, we directly parameterize the policy over the displacement vector between noise and clean actions. This displacement, illustrated as the red vector in Figure 2 (1), represents the full transport from noise to data. Crucially, this global displacement admits a recursive decomposition into shorter displacements of the same form, naturally inducing a divide-and-conquer structure. As shown in Figure 2 (1), a long-range displacement (red) can be obtained by composing intermediate displacements (yellow), each of which can in turn be decomposed into shorter displacements (blue). This hierarchical structure admits a well-defined boundary condition: as the interval length shrinks, the displacement converges to the marginal velocity multiplied by the time increment. This limiting case can be solved using standard flow matching, providing a principled base case and a natural stopping criterion for the recursive decomposition.

Motivated by this structure, we employ a single shared network to model displacements at all transition scales. Training proceeds by first learning short-range displacements near the boundary case, and then progressively aggregating these learned components to model longer-range displacements, ultimately recovering the full noise-to-action mapping in a single step.

### 4.1. Policy Modeling and Behavior Cloning Objective

We begin by modeling the policy following the intuition described above, before integrating it into a behavior-regularized actor-critic framework to obtain the complete method. In this part, our goal is to construct a policy that can directly map a noisy action to a clean action in a single step while remaining consistent with the underlying continuous-time generative dynamics. We start from the standard Flow Matching formulation, where actions evolve according to a continuous velocity field, and derive a compositional structure that enables efficient long-range transitions. We then introduce boundary constraints that anchor the learned policy to the local flow dynamics and prevent degenerate solutions. Finally, we formulate the behavior cloning objective, which serves as the core training loss used in the behavior-regularized actor-critic framework.

Mathematically, we consider a continuous-time generative process defined over $t \in [0, 1]$, where $a_t \in \mathbb{R}^d$ denotes the noising action at time $t$. The process interpolates between clean action at $t = 0$ and full noise at $t = 1$ for a given state $s$ and is governed by an underlying marginal velocity field $v(a_t, t; s)$:

$$\frac{da_t}{dt} = v(a_t, t; s). \tag{9}$$

Rather than directly modeling the marginal velocity, we focus on the *displacement* over a finite interval $[r, t]$, which plays a central role in efficient transitions.

$$d(a_t, r, t; s) \triangleq \int_r^t v(a_\tau, \tau; s) \, d\tau. \tag{10}$$

Define the *policy operator* that directly maps a noisy action at time $t$ to a less-noisy action at an earlier time $r$:

$$\pi(a_t, r, t; s) \triangleq a_t - d(a_t, r, t; s). \tag{11}$$

Under the true dynamics, this policy satisfies $\pi(a_t, r, t; s) = a_r$.

**Compositional Consistency.** A fundamental property of the displacement follows from the additivity of definite integrals. For any $0 \le r \le m \le t \le 1$,

$$
\begin{aligned}
d(a_t, r, t; s) &= \int_r^t v(a_\tau, \tau; s) \, d\tau \\
&= \int_r^m v(a_\tau, \tau; s) \, d\tau + \int_m^t v(a_\tau, \tau; s) \, d\tau \\
&= d(a_m, r, m; s) + d(a_t, m, t; s).
\end{aligned}
\tag{12}
$$

Rewriting this identity in terms of the policy operator yields the following compact compositional constraint:

$$
\begin{aligned}
\pi(a_t, r, t; s) = \pi(\pi(a_t, m, t; s), \, r, \, m; s), \\
\forall \, 0 \le r \le m \le t \le 1.
\end{aligned}
\tag{13}
$$

This equation states that a direct transition from $t$ to $r$ must be equivalent to a two-step transition from $t$ to $m$ followed by $m$ to $r$.

**Boundary Conditions.** To prevent degenerate solutions and anchor learning to the true local dynamics of the flow matching, we impose explicit boundary conditions on the policy operator $\pi$.

For a zero-length interval, the policy must reduce to the identity map:

$$\pi(a_t, t, t; s) = a_t. \tag{14}$$

For an infinitesimal time interval $\Delta > 0$, the policy should be consistent with the marginal velocity field:

$$\pi(a_t, t - \Delta, t; s) \approx a_t - \Delta * v(a_t, t; s). \tag{15}$$

In practice, directly enforcing this constraint becomes difficult when $\Delta \to 0$. In this regime, the displacement term $\Delta * v_\theta(\mathbf{x}_t, t, \mathbf{s})$ becomes extremely small, causing

the marginal velocity information—which provides the key local dynamics we want to anchor the policy to—to be carried only through a vanishingly small update. As a result, the training signal becomes weak, leading to unstable and ineffective optimization when directly learning the displacement.

To address this issue, we adopt a lightweight architectural preconditioning: the policy network is parameterized to predict a velocity-like quantity, which is then used to obtain the action update $\pi_\theta(a_t, t - \Delta, t; s) \approx a_t - \Delta * v_\theta(a_t, t; s)$. This design is illustrated in Figure 2 (2).

With this parameterization, for sufficiently small $\Delta$, the policy can directly query an explicit velocity estimate and anchor itself to the local flow dynamics, where such local information is most important. For larger $\Delta$, this constraint is unnecessary, allowing the policy to retain sufficient flexibility to model non-local transitions. Furthermore, as $\Delta \to 0$, the identity condition in Eq. 14 is automatically satisfied.

**Boundary Condition via Conditional Velocity.** The boundary constraint above requires accurate estimates of the marginal velocity field. A straightforward approach would be to pretrain a separate flow model and use its predicted velocity as a supervision signal. However, this introduces additional training stages and model complexity. Instead, we directly exploit the conditional velocity $v_{\text{cond}}$ naturally provided by Flow Matching as a built-in supervision signal. This yields a simple and efficient single-stage, single-model training objective without requiring a pretrained teacher model.

**Behavior Cloning Objective.** Let $\pi_\theta$ denote a neural parameterization of the policy, and let $v_\theta$ denote its intermediate velocity-like prediction, which can be queried when enforcing boundary constraints. Given a sampled point $a_t$ and times $0 \le r \le m \le t \le 1$, we define the compositional consistency loss as:

$$\mathcal{L}_{\text{comp}}(\theta) = \mathbb{E}_{\mathcal{D}, a_t, \, r \le m \le t}\left[\left\|\pi_\theta(a_t, r, t; s) - \text{sg}(\tilde{a}_r)\right\|_2^2\right],$$
$$\tilde{a}_r = \pi_\theta(\pi_\theta(a_t, m, t; s), \, r, \, m; s).$$
(16)

where $\text{sg}(\cdot)$ denotes the stop-gradient operator.

For boundary cases corresponding to small $\Delta$, we additionally impose:

$$\mathbb{E}\left[\left\|\pi_\theta(a_t, t - \Delta, t; s) - \text{sg}(a_t - \Delta * v(a_t, t; s))\right\|_2^2\right].$$
(17)

Using the proposed parameterized architecture, this objective simplifies to a numerically stable, on-the-fly velocity matching condition:

$$\mathcal{L}_{\text{bnd}}(\theta) = \mathbb{E}_{\mathcal{D}}\left[\left\|v_\theta(a_t, t - \Delta, t; s) - \text{sg}(v_{\text{cond}}(a_t, t \mid a, \epsilon; s))\right\|_2^2\right]$$
(18)

which directly aligns the learned velocity with the conditional Flow Matching velocity at the boundary.

The behavior cloning objective is defined as

$$\mathcal{L}_{BC}(\theta) = (1 - \lambda)\,\mathcal{L}_{\text{comp}} + \lambda\,\mathcal{L}_{\text{bnd}},$$
(19)

where $\lambda \in [0, 1]$ is a fixed hyperparameter controlling the boundary conditioning ratio.

For efficiency, instead of computing both losses at every iteration, we sample

$$\xi \sim \text{Bernoulli}(\lambda),$$
(20)

and optimize the stochastic objective

$$\hat{\mathcal{L}}_{BC}(\theta) = (1 - \xi)\,\mathcal{L}_{\text{comp}} + \xi\,\mathcal{L}_{\text{bnd}}.$$
(21)

This provides an unbiased estimator of Eq. (19), i.e.,

$$\mathbb{E}_\xi\left[\hat{\mathcal{L}}_{BC}(\theta)\right] = \mathcal{L}_{BC}(\theta).$$
(22)

## 4.2. Bootstrapped Flow Q-Learning.

We construct *Bootstrapped Flow Q-Learning (BFQ)* by integrating the proposed flow-based policy parameterization and behavior cloning objective into a behavior-regularized actor–critic framework as detailed in Section 3. The behavior regularization term is given by $\mathcal{L}_{\text{BC}}(\theta)$, as defined in Eq. 19.

Given the policy $\pi_\theta$, action sampling reduces to a differentiable single-step operation:

$$a = \pi_\theta(\epsilon, r = 0, t = 1; s), \qquad \epsilon \sim \mathcal{N}(0, I), \quad (23)$$

which enables efficient action generation without iterative denoising.

The critic and actor are optimized using the following objectives:

$$\mathcal{L}_Q(\phi) = \mathbb{E}_{\mathcal{D}, a' \sim \pi_{\theta'}}\left[\left(r + \gamma \min_i Q_{\phi'_i}(s', a') - Q_{\phi_i}(s, a)\right)^2\right],$$
(24)
$$\mathcal{L}(\theta) = \mathcal{L}_{\text{BC}}(\theta) - \alpha\,\mathbb{E}_{s \sim \mathcal{D},\, a^\pi \sim \pi_\theta}\left[Q_\phi(s, a^\pi)\right].$$
(25)

To normalize across dataset-specific Q-value scales, following (Fujimoto & Gu, 2021; Wang et al., 2022), the coefficient $\alpha$ is further adapted as:

$$\alpha = \frac{\eta}{\mathbb{E}_{(s,a) \sim \mathcal{D}}\left[|Q_\phi(s, a)|\right]},$$
(26)

where $\eta$ is a tunable hyperparameter. The normalization term is treated as a constant with respect to gradient updates.

For more details on the complete training procedure of BFQ, please refer to Appendix E.

| Policy Type | Gaussian Policy | | | Diffusion Policy | | | One-Step Flow/Diffusion Policy | | | | |
|---|---|---|---|---|---|---|---|---|---|---|---|
| Dataset | BC | TD3-BC | IQL | EDP | IDQL | DQL | SRPO | SORL | OFQL | FQL | BFQ (Ours) |
| HalfCheetah-Medium-Expert | 60.2±13.2 | 91.5±15.8 | 88.3±2.8 | 95.8±0.1 | 91.3±0.6 | 95.5±0.1 | 92.2±3.0 | 96.5±0.9 | 95.2±0.4 | **99.8±0.1** | 98.6±0.1 |
| Hopper-Medium-Expert | 67.2±20.6 | 101.6±23.2 | 76.6±34.9 | 110.8±0.4 | 110.1±0.7 | **111.1±0.4** | 100.1±13.9 | 45.9±6.7 | 110.2±1.3 | 86.2±1.3 | 110.5±1.5 |
| Walker2d-Medium-Expert | 105.4±3.1 | 110.4±0.4 | 108.7±2.2 | 110.4±0.0 | 110.6±0.0 | 111.6±0.9 | 114.0±2.1 | 109.1±1.7 | 113.0±0.1 | 100.5±0.1 | **113.4±0.1** |
| HalfCheetah-Medium | 42.5±0.2 | 48.5±0.7 | 47.7±0.2 | 50.8±0.0 | 51.5±0.1 | 52.3±0.2 | 60.4±0.8 | 57.4±0.7 | 63.8±0.1 | 60.1±0.1 | **66.1±0.0** |
| Hopper-Medium | 52.9±0.2 | 56.6±9.0 | 61.2±6.4 | 72.6±0.2 | 70.1±2.0 | 96.5±1.3 | 95.5±2.0 | 81.3±5.8 | **103.6±0.1** | 74.5±0.2 | 103.5±0.1 |
| Walker2d-Medium | 72.3±3.2 | 83.3±7.0 | 78.7±4.5 | 86.5±0.2 | 86.1±0.4 | 86.8±0.0 | 84.4±4.4 | 77.9±2.5 | 87.4±0.1 | 72.7±0.8 | **91.7±0.1** |
| HalfCheetah-Medium-Replay | 34.7±1.8 | 44.5±0.8 | 42.9±1.7 | 44.9±0.4 | 46.5±0.3 | 47.9±0.0 | 51.4±3.4 | 48.3±0.2 | 51.2±0.1 | 51.1±0.1 | **52.1±0.1** |
| Hopper-Medium-Replay | 19.7±5.5 | 55.2±24.6 | 86.8±15.5 | 83.0±1.7 | 99.4±0.1 | 91.2±0.0 | 101.2±1.0 | 93.0±2.1 | 101.9±0.7 | 85.4±0.5 | **102.1±0.1** |
| Walker2d-Medium-Replay | 29.0±0.5 | 82.5±13.6 | 68.3±6.4 | 87.0±2.6 | 89.1±2.4 | 98.2±0.1 | 84.6±7.1 | 67.8±11.8 | **106.2±0.6** | 82.1±1.2 | 97.5±1.2 |
| **Average (MuJoCo)** | 53.8 | 74.9 | 73.2 | 82.4 | 84.1 | 89.0 | 87.1 | 80.0 | 92.5 | 79.2 | **92.8** |
| AntMaze-Medium-Play | 0.0±0.0 | 10.6±10.1 | 75.5±5.5 | 73.3±6.2 | 67.3±5.7 | 86.0±1.8 | 80.7±7.1 | 80.1±11.6 | **88.1±5.0** | 78.0±7.0 | 87.0±4.0 |
| AntMaze-Large-Play | 0.0±0.0 | 0.2±0.4 | 38.6±4.2 | 33.3±1.9 | 48.7±4.7 | 83.3±2.5 | 53.6±12.5 | 57.3±16.4 | 84.0±6.1 | 84.0±7.0 | **88.5±4.8** |
| AntMaze-Medium-Diverse | 0.6±0.2 | 5.7±8.2 | 78.0±1.5 | 52.7±1.9 | 83.3±5.0 | 94.7±2.5 | 75.0±12.3 | 70.0±8.1 | **90.2±4.2** | 71.0±13.0 | 84.1±6.9 |
| AntMaze-Large-Diverse | 0.0±0.0 | 0.0±0.0 | 48.0±7.5 | 41.3±3.4 | 40.0±11.4 | 61.3±8.4 | 53.6±6.3 | 52.5±10.9 | 76.1±6.6 | **83.0±4.0** | 76.1±7.0 |
| **Average (AntMaze)** | 0.2 | 4.1 | 60.0 | 50.2 | 59.8 | 81.3 | 73.6 | 65.0 | **84.6** | 79.0 | 83.9 |

*Table 1.* Comparison of normalized scores on D4RL benchmarks across MuJoCo and AntMaze domains. Bold values indicate the best performance in each row.

# 5. Experimental setting

**Benchmarks.** We evaluate BFQ on a diverse set of tasks from the D4RL benchmark suite (Fu et al., 2020), a widely adopted standard for offline RL. Our evaluation spans various domains, including locomotion, navigation and manipulation tasks to demonstrate the method's generalizability. Detailed task descriptions and experimental protocols are provided in Appendix A.

**Baselines.** To rigorously assess BFQ's performance, we compare it against a broad spectrum of representative baselines, categorized as follows: (1) Non-Diffusion policies: Behavior Cloning (BC), TD3-BC (Fujimoto & Gu, 2021), and IQL (Kostrikov et al., 2021); (2) Multi-step Diffusion-based policies: IDQL (Hansen-Estruch et al., 2023), DQL (Wang et al., 2022), and EDP (Kang et al., 2023); and (3) single-step Flow policies: SRPO (Chen et al., 2023), SORL (Espinosa-Dice et al., 2026), OFQL (Nguyen & Yoo, 2026), FQL (Park et al., 2025).

**Implementation Details.** Unless otherwise specified, our implementation is built on top of CleanDiffuser (Dong et al., 2024), following its evaluation protocol to ensure fair and consistent comparison with prior diffusion-based methods. We adopt conventional MLP architectures for both the policy and Q-functions, with detailed hyperparameter settings provided in Appendix H. The minor difference is that the policy input is formed by concatenating the action latent vector, the current state vector, and sinusoidal positional embeddings of timesteps $t$ and $r$ (embedding dimension 64).

For timestep sampling, $t$, $r$, and $m$ are uniformly sampled subject to $0 \le r < m < t \le 1$. The boundary-condition offset $\Delta$ is sampled from $[0, \Delta_{\max}]$, and we empirically find that $\Delta_{\max} = 10^{-3}$ provides robust performance across all environments.

| Method | Steps | Frequency | Training Time |
|---|---|---|---|
| BFQ | 1 | 851.2 | **7.8** |
| FQL | 1 | **929.6** | NA |
| | 5 | NA | 7.9 |
| | 10 | NA | 8.5 |
| | 20 | NA | 9.3 |
| | 50 | NA | 13.5 |
| DQL | 5 | 238.1 | 11.7 |
| | 10 | 150.1 | 16.1 |
| | 20 | 75.2 | 24.8 |
| | 50 | 35.5 | 49.5 |

*Table 2.* Comparison of action frequency (Hz) and training wall-clock time (hours) over one million training steps, averaged across MuJoCo tasks, for different methods and sampling steps. Action frequency for FQL is measured using its single-step inference model, while training time for FQL corresponds to the number of FM steps used during distillation.

The main hyperparameters of BFQ are the *boundary conditioning ratio* $\lambda$ and the coefficient $\eta$. Unless otherwise specified, we fix $\lambda = 0.5$ and tune $\eta$ via grid search over $\{0.001, 0.01, 0.05, 0.1, 0.3, 0.5, 1\}$. We use the Adam optimizer with a learning rate of $3 \times 10^{-4}$.

To ensure reliable evaluation, we report BFQ performance using the average D4RL normalized score (Fu et al., 2020) over six random seeds, where each trained policy is evaluated on 50 episodes, resulting in 300 total evaluation episodes per reported result. Task-specific hyperparameter settings are provided in Appendix H.

# 6. Experimental result

Benchmark results are summarized in Table 1, with detailed discussion provided below.

**Locomotion Domain (MuJoCo).** BFQ demonstrates strong and consistent performance across MuJoCo locomotion

tasks, outperforming most prior diffusion-based and one-step policy baselines. In particular, BFQ achieves the best average performance among all compared methods, improving the average score from 89.0 for DQL to 92.8. The improvements are especially pronounced on medium and medium-replay datasets, which contain noisier and more suboptimal trajectories that typically induce highly multimodal action distributions. These results highlight the importance of expressive policy representations together with stable value estimation.

Compared with existing one-step policy approaches such as SRPO, SORL, and FQL, BFQ achieves substantially stronger and more consistent performance across datasets. While methods such as EDP and IDQL improve efficiency by simplifying diffusion sampling or mitigating BPTT, they often sacrifice final performance relative to DQL. In contrast, BFQ preserves the advantages of expressive policy learning while avoiding BPTT during Q-learning, leading to improved optimization stability and stronger overall results. Notably, BFQ also consistently outperforms FQL on most locomotion tasks, despite both methods employing one-step generation.

**AntMaze Domain.** AntMaze tasks are particularly challenging due to sparse rewards, long-horizon credit assignment, and highly suboptimal demonstrations, making stable Q-learning crucial for effective policy optimization. As expected, methods without explicit Q-learning signals, such as BC, perform poorly, whereas Q-learning–based approaches including DQL, SRPO, OFQL, and BFQ achieve substantially stronger performance.

Among all compared methods, BFQ achieves highly competitive performance and attains the best results on the challenging AntMaze-Large-Play benchmark. BFQ achieves an average score of 83.9, outperforming most diffusion-based and one-step baselines while remaining competitive with the strongest prior method, OFQL (84.6). Compared to FQL, BFQ consistently improves performance across most AntMaze tasks, suggesting that boundary-conditioned flow matching provides more stable and effective policy optimization under sparse-reward settings. Overall, these results demonstrate that BFQ effectively combines the efficiency advantages of one-step policy generation with strong value-guided policy learning.

## 7. Ablation Study

| $\lambda$ | 1 | 0.75 | 0.5 | 0.25 | 0 |
|---|---|---|---|---|---|
| Medium Expert | 80.9 | 96.1 | **98.5** | 94.0 | -2.5 |
| Medium | 45.3 | 63.7 | **66.1** | 62.1 | -2.5 |
| Medium Replay | 49.0 | 50.7 | **52.1** | 51.8 | 10.6 |

*Table 3.* Ablation on boundary conditioning ratio on HalfCheetah

**Training and Inference Efficiency Comparison.** Figure 2

summarizes wall-clock training time (1M steps) and decision frequency (Lu et al., 2025b) measured on an A100 GPU (Appendix D). DQL shows nearly linear growth in training cost as the number of denoising steps increases, rising from 11.7 hours at 5 steps to 49.5 hours at 50 steps, whereas BFQ completes training in 7.8 hours. At inference, BFQ achieves a decision frequency of 851.2 Hz, substantially exceeding both 5-step DQL (238.7 Hz) and 50-step DQL (35.5 Hz).

Compared to the single-step FQL baseline, BFQ attains comparable inference speed but requires less training time, as FQL depends on multiple NFEs to construct distillation targets. Despite similar efficiency, FQL consistently lags behind BFQ in policy performance. Overall, BFQ provides faster training and higher decision frequency while maintaining strong policy expressiveness.

**On Boundary Conditioning Ratio**. We examine the impact of the boundary conditioning ratio $\lambda$ on policy performance across HalfCheetah datasets (Table 3). The best results are achieved at $\lambda = 0.5$, yielding scores of 98.5 on Medium Expert, 66.1 on Medium, and 52.1 on Medium Replay. In contrast, extreme settings ($\lambda = 0$ or $\lambda = 1$) lead to noticeable performance degradation; in particular, removing boundary conditioning entirely ($\lambda = 0$) prevents meaningful policy learning. These results suggest that a moderate boundary conditioning ratio acts as an effective regularizer, promoting stable and robust learning.

**On the expressiveness of FM and BFQ policies.** Prior work on diffusion-based Q-learning (DQL) suggests that increased policy expressiveness improves performance in actor–critic frameworks. To evaluate expressiveness under single-step generation, we compare FM and BFQ policies on a toy dataset (Appendix C). Figure 3 shows that FM requires multiple generation steps to achieve good coverage and exhibits mode collapse when evaluated with fewer steps. In contrast,the BFQ policy achieves strong mode coverage and closely matches the target distribution in a single step.

**On the Value Guidance Hyperparameter** $\eta$

The value guidance hyperparameter $\eta$ controls the strength of value guidance relative to the behavior cloning objective during optimization. Similar to prior offline RL methods, the optimal choice of $\eta$ depends on the suboptimality of the dataset. To analyze its effect, we perform an ablation study over different $\eta$ values on the HalfCheetah benchmark, as shown in Table 4.

Overall, BFQ is relatively robust to the choice of $\eta$, achieving strong performance across a broad range of values. We observe that larger $\eta$ values are generally more effective for lower-quality and more suboptimal datasets (e.g., Medium and Medium-Replay), whereas smaller $\eta$ values are preferred for higher-quality datasets such as Medium-Expert. This behavior is consistent with the intuition that stronger

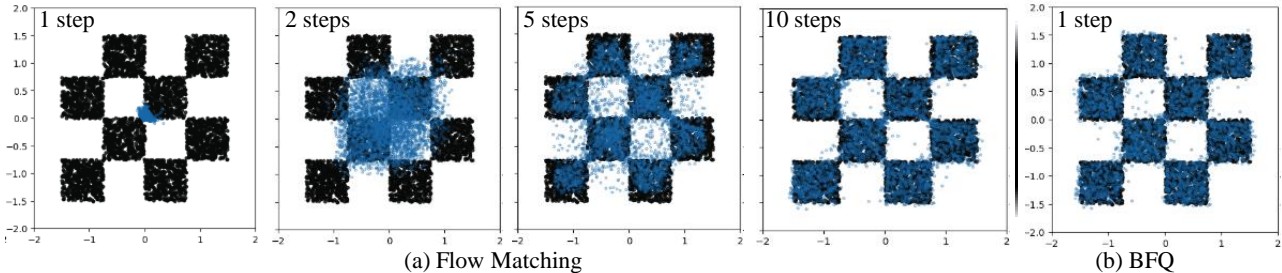

(a) Flow Matching                                                                                      (b) BFQ

*Figure 3.* Qualitative comparison of bandit action distribution modeling on a toy dataset with complex multimodal structure, contrasting FM (left, evaluated with up to 10 generation steps) and BFQ (Ours) (right, achieving comparable modeling with a single generation step)

| $\eta$ | 0.01 | 0.05 | 0.1 | 0.5 | 1 | 5 | 10 |
|---|---|---|---|---|---|---|---|
| HalfCheetah-Medium-Expert | **95.1** ± 0.2 | **98.5** ± 0.1 | **95.6** ± 0.1 | 69.7 ± 3.5 | 68.2 ± 1.8 | 56.5 ± 4.2 | 47.7 ± 3.5 |
| HalfCheetah-Medium | 50.1 ± 0.05 | 58.1 ± 0.08 | **60.1** ± 0.1 | 65.1 ± 0.1 | **66.1** ± 0.0 | 63.6 ± 0.2 | 59.1 ± 0.2 |
| HalfCheetah-Medium-Replay | 44.3 ± 0.1 | 46.7 ± 0.1 | **48.5** ± 0.1 | 51.1 ± 0.1 | **52.1** ± 0.1 | 51.1 ± 0.2 | 49.1 ± 1.1 |

*Table 4.* Ablation study on the normalization coefficient $\eta$ across D4RL HalfCheetah tasks: Medium-Expert, Medium, and Medium-Replay. Results are reported as mean ± standard deviation. We highlight the best score and those achieving at least 90% of the maximum performance.

value guidance becomes increasingly beneficial when the dataset contains a larger proportion of suboptimal trajectories.

## 8. Conclusion

In this paper, we introduced Bootstrapped Flow Q-Learning (BFQ), a simple yet effective framework for offline RL that enables accurate single-step action generation during both training and inference. BFQ departs from prior diffusion- and flow-based policies by directly learning the noise-to-action displacement through a divide-and-conquer bootstrapping strategy, thereby eliminating multi-step denoising and backpropagation through time (BPTT) without relying on auxiliary models, policy distillation. Experiments on standard D4RL benchmarks show that BFQ achieves strong or state-of-the-art performance while substantially reducing computational cost. This work highlights the potential of single-step flow-based policies as a practical and scalable alternative to diffusion-based approaches, opening new directions for efficient policy learning in offline and resource-constrained settings.

**Limitation.** This work focuses on state-based D4RL tasks in offline RL, as they remove confounding factors such as visual feature extraction, multimodal fusion, and architecture-specific biases. This setting allows us to more precisely isolate and analyze the algorithmic contributions of BFQ. Extending BFQ to more complex settings—such as offline-to-online learning or visual observation–based tasks—is a natural direction for future work and remains beyond the scope of this paper.

## Acknowledgments

This work was supported by the Institute for Information & Communications Technology Planning & Evaluation (IITP) grant funded by the Korea government (MSIT) under Grant No. RS-2021-II211381, "Development of Causal AI through Video Understanding and Reinforcement Learning, and Its Applications to Real Environments," and by the National Research Foundation of Korea (NRF) grant funded by the Korea government (MSIT) under Grant No. RS-2025-24742969, "Intelligent Robotic System using Continual Learning and Multimodal Language Model based Multi-Attribute Feedback."

## Impact Statement

This paper presents methodological advances in offline RL, with the goal of improving the efficiency and practicality of generative policy learning. By reducing the computational cost and complexity of policy inference, the proposed approach may facilitate wider adoption of offline reinforcement learning in domains where online interaction is costly or unsafe, such as robotics, healthcare, and autonomous systems.

As with many RL methods, potential risks arise if learned policies are deployed in real-world systems without appropriate validation, oversight, or safety constraints. However, this work does not introduce new capabilities that fundamentally alter existing risk profiles beyond those already present in the offline RL literature. We do not anticipate immediate negative societal consequences specific to the proposed method. Overall, we believe this work contributes positively to the development of more efficient and accessible RL techniques, while highlighting the continued importance of responsible evaluation and deployment practices.

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

| Task | Gaussian Policies | | Diffusion Policies | | Flow Policies | | | | One-Step Diffusion/Flow Policy | | | |
|---|---|---|---|---|---|---|---|---|---|---|---|---|
| | BC-1 | IQL-1 | IDQL-5 | CAC-2 | FAWAC-10 | FBRAC-10 | IFQL-10 | SORL-8 | SORL-1 | SRPO-1 | FQL-1 | BFQ-1 |
| antmaze-large-navigate-singletask-v0 | $0 \pm 0$ | $48 \pm 9$ | $0$ | $42 \pm 7$ | $1 \pm 1$ | $70 \pm 20$ | $24 \pm 17$ | $93 \pm 2$ | $\mathbf{93} \pm 2$ | $0 \pm 0$ | $80 \pm 8$ | $91 \pm 2$ |
| antmaze-giant-navigate-singletask-v0 | $0 \pm 0$ | $0 \pm 0$ | $0 \pm 0$ | $0 \pm 0$ | $0 \pm 0$ | $0 \pm 1$ | $0 \pm 0$ | $\mathbf{12} \pm 6$ | $0 \pm 0$ | $0 \pm 0$ | $4 \pm 5$ | $0 \pm 0$ |
| humanoidmaze-medium-navigate-singletask-v0 | $1 \pm 0$ | $32 \pm 7$ | $1 \pm 1$ | $38 \pm 19$ | $6 \pm 2$ | $25 \pm 8$ | $69 \pm 19$ | $67 \pm 4$ | $80 \pm 8$ | $0 \pm 0$ | $19 \pm 12$ | $\mathbf{94} \pm 3$ |
| humanoidmaze-large-navigate-singletask-v0 | $0 \pm 0$ | $3 \pm 1$ | $0 \pm 0$ | $1 \pm 1$ | $0 \pm 0$ | $0 \pm 1$ | $6 \pm 2$ | $20 \pm 9$ | $0 \pm 0$ | $0 \pm 0$ | $7 \pm 6$ | $\mathbf{48} \pm 5$ |
| cube-single-play-singletask-v0 | $3 \pm 1$ | $85 \pm 8$ | $96 \pm 2$ | $80 \pm 20$ | $12 \pm 3$ | $24 \pm 4$ | $16 \pm 9$ | $\mathbf{99} \pm 0$ | $82 \pm 6$ | $0 \pm 0$ | $97 \pm 2$ | $90 \pm 5$ |
| scene-play-singletask-v0 | $0 \pm 0$ | $12 \pm 3$ | $33 \pm 14$ | $50 \pm 40$ | $18 \pm 8$ | $46 \pm 10$ | $0 \pm 0$ | $\mathbf{89} \pm 9$ | $1 \pm 1$ | $2 \pm 2$ | $76 \pm 9$ | $82 \pm 5$ |
| Average (OGBench) | 1 | 30 | 22 | 35 | 6 | 28 | 19 | 63 | 43 | 0 | 47 | **68** |
| pen-human-v1 | $72 \pm 5$ | $75 \pm 8$ | $76 \pm 10$ | $64 \pm 8$ | $67 \pm 5$ | $77 \pm 7$ | $71 \pm 12$ | $70 \pm 3$ | $64.5 \pm 5$ | $69 \pm 7$ | $53 \pm 6$ | $\mathbf{82} \pm 7$ |
| pen-cloned-v1 | $51 \pm 7$ | $73 \pm 10$ | $64 \pm 7$ | $56 \pm 10$ | $62 \pm 10$ | $67 \pm 9$ | $\mathbf{80} \pm 11$ | $75 \pm 11$ | $74.2 \pm 7$ | $61 \pm 7$ | $74 \pm 11$ | $75 \pm 7$ |
| Average (Adroit) | 62 | 74 | 70 | 60 | 65 | 72 | 76 | 73 | 69 | 65 | 64 | **79** |

*Table 5.* Results on OGBench, covering six representative tasks across diverse domains: navigation (AntMaze-Large and AntMaze-Giant), high-dimensional locomotion (HumanoidMaze-Medium and HumanoidMaze-Large), and robot manipulation (Cube-Single and Scene-Play). We further evaluate performance on D4RL Adroit tasks (Pen-Human and Pen-Cloned). *Note:* We follow the default task selection recommended by the OGBench authors. In addition, we adopt the experimental settings and reporting conventions of FQL, including its recommended hyperparameters and baseline configurations. CAC-2 denotes CAC with 2 denoising steps, while SORL-8 refers to the variant using 8 denoising steps. Further details on baselines and hyperparameters can be found in Appendix E.2 of FQL and SORL.

## A. Benchmarking tasks and evaluation protocol

**Benchmarking tasks**. We evaluate BFQ on a diverse set of benchmark tasks spanning multiple reinforcement learning domains. These tasks are selected to assess BFQ's robustness and generalization across varied environment dynamics and decision-making horizons. Specifically, our evaluation includes locomotion tasks that emphasize short-horizon control, and navigation tasks focused on goal-directed pathfinding. This diverse task suite enables a comprehensive assessment of BFQ's performance across both simple and challenging settings, providing insight into its strengths and limitations.

Locomotion (MuJoCo). MuJoCo locomotion tasks constitute a standard benchmark in reinforcement learning, where agents control simulated robots under complex physical dynamics. These environments primarily evaluate the agent's ability to execute fine-grained control and short-term decision-making while maintaining stability and efficiency in continuous control settings.

Navigation (AntMaze). AntMaze tasks integrate continuous control with long-horizon planning in maze-like environments. Agents must navigate through increasingly complex maze layouts, requiring effective coordination between locomotion and strategic planning. This benchmark evaluates the agent's ability to balance exploration and exploitation while pursuing goal-directed behavior under sparse reward signals.

**Evaluation Metric.** We follow the D4RL benchmark (Fu et al., 2020) and report performance using the normalized score, which facilitates fair comparisons across different algorithms and environments. For each task, the normalized score is computed as

$$\text{Normalized Score} = 100 \times \frac{\text{score} - \text{random score}}{\text{expert score} - \text{random score}}. \tag{27}$$

A normalized score of 0 corresponds to the average return of a uniformly random policy evaluated over 100 episodes, while a score of 100 reflects the average return achieved by a task-specific expert policy defined in D4RL.

## B. Extending empirical studies beyond D4RL MuJoCo locomotion

We conduct experiments on OGBench, selecting six representative tasks spanning diverse domains, including navigation (AntMaze-Large and AntMaze-Giant), high-dimensional locomotion (HumanoidMaze-Medium and HumanoidMaze-Large), and robotic manipulation (Cube-Single and Scene-Play). We further evaluate our method on the D4RL Adroit benchmark using the pen-human and pen-cloned tasks. Our implementation is built upon the framework of (Wang et al., 2026), adopting its policy and Q-network architectures. The results are summarized in Table 5.

On OGBench, BFQ achieves strong performance across all task categories despite using a single-step policy. It attains the highest average score (68), outperforming strong baselines such as SORL-8 (63) and FQL (47). BFQ also achieves top or near-top results on most tasks (e.g., 94 on humanoidmaze-medium and 91 on antmaze-large), demonstrating that it can match or surpass multi-step methods while offering one-step inference.

On the D4RL Adroit benchmark, BFQ shows strong and consistent performance across both tasks, achieving the highest

average score (79). It obtains the best result on pen-human-v1 (82 ± 7), surpassing FBRAC (77 ± 7) and IDQL (76 ± 10), and remains competitive on pen-cloned-v1 (75 ± 7), close to IFQL (80 ± 11) while maintaining a simpler one-step policy.

## C. Flow Matching and BFQ Policies on Toy Datasets

**Experiment Setup.** Motivated by the observation that increased policy expressiveness improves actor–critic performance (Wang et al., 2022), we study whether complex action distributions can be modeled accurately in a single-step setting. We compare (i) a Flow Matching (FM) policy and (ii) a BFQ policy on a synthetic checkerboard dataset, where points are arranged in a multimodal checkerboard pattern to test distributional expressiveness.

**Architecture.** The FM policy predicts marginal velocities using an MLP with three hidden layers of 64 units, taking the noise latent and timestep as inputs. The BFQ policy uses a similar MLP architecture and preconditioning, but directly predicts actions with an additional target-time input.

**Training and Evaluation.** Both models are trained for 100 epochs with batch size 2048 and 40 batches per epoch. The FM policy is evaluated using varying numbers of prediction steps (1, 2, 5, and 10), while BFQ is evaluated using a single step. We visualize ground-truth samples (black) and generated samples (blue) in 2D to assess how well each method captures the underlying geometric structure.

## D. Training and Inference Efficiency Comparison

We evaluate both decision frequency and training efficiency of our method and baseline approaches across nine MuJoCo tasks. Following Lu et al. (2025b), decision frequency measures the number of actions (or action batches) generated per second during inference.

All experiments are conducted on an Ubuntu server equipped with an Intel(R) Xeon(R) Gold 5317 CPU (48 cores, 96 threads) and an NVIDIA A100 PCIe GPU with 80 GB memory. Training efficiency is reported as wall-clock time (hours) measured over one million training steps and averaged across the nine tasks. Decision frequency is computed by averaging performance over 3,000 action batches (batch size 2,500) per task across all tasks.

## E. Pseudocode for Bootstrapped Flow Q-Learning (BFQ)

The complete training procedure of BFQ is summarized in Algorithm 1. In addition, the detailed computation of the behavior cloning objective, including the boundary anchoring and compositional consistency mechanisms, is provided in Algorithm 2.

---

**Algorithm 1** BFQ Algorithm

---

1: **Initialize** policy network $\pi_\theta$ , $\pi_{\theta'}$, critic networks $Q_{\phi_1}$ and $Q_{\phi_2}$, $Q_{\phi'_1}$, $Q_{\phi'_2}$
2: **for** each iteration **do**
3:      Sample transition mini-batch $\mathcal{B} = \{(s_h, a_h, r_h, s_{h+1})\} \sim \mathcal{D}$
4:      **# Q-value function learning**
5:      Sample $a_{h+1} \sim \pi_{\theta'}(a_{h+1} \mid s_{h+1})$ by Eq. 23
6:      Update $Q_{\phi_1}$ and $Q_{\phi_2}$ using Eq. 24 {Max-Q backup optional}
7:      **# Policy learning**
8:      Sample $a_h \sim \pi_\theta(a_h \mid s_h)$ by Eq. 23
9:      Update policy $\pi_\theta$ by minimizing Eq. 25
10:      **# Update target networks every K iteration**
11:      $\theta' \leftarrow \rho\theta' + (1 - \rho)\theta$
12:      $\phi'_i \leftarrow \rho\phi'_i + (1 - \rho)\phi_i$ for $i = \{1, 2\}$
13: **end for**

---

---

**Algorithm 2** Compute Behavior Cloning Loss

---

**Input:** policy $\pi_\theta$, probability $\lambda$, maximum step $\Delta_{\max}$
Sample data $(a, s) \sim \mathcal{D}$ and noise $\epsilon \sim \mathcal{N}(0, I)$
Sample time $t \sim \mathcal{U}(0, 1)$
Construct noisy action $a_t \leftarrow (1 - t)a + t\epsilon$
Compute conditional velocity $v_{\text{cond}} \leftarrow \epsilon - a$
Sample $u \sim \mathcal{U}(0, 1)$
**if** $u < \lambda$ **then**
   # Boundary anchoring
   Sample $\Delta \sim \mathcal{U}(0, \Delta_{\max})$ and set $r \leftarrow t - \Delta$
   $\mathcal{L} \leftarrow \mathcal{L}_{\text{bnd}}(a_t, r, t)$                                                 // Eq. (18)
**else**
   # Compositional consistency
   Sample $r \sim \mathcal{U}(0, t - \Delta_{\max})$ and $m \sim \mathcal{U}(r, t)$
   $\mathcal{L} \leftarrow \mathcal{L}_{\text{comp}}(a_t, r, m, t)$                                          // Eq. (16)
**end if**
**Output:** loss $\mathcal{L}_{BC}$

---

## F. Necessity of the Preconditioning Mechanism

**Why is velocity-field parameterization needed?** Although BFQ is conceptually formulated as learning a direct flow map (i.e., displacement prediction), the displacement itself corresponds to the time integral of an underlying marginal velocity field. Importantly, this velocity field cannot be arbitrary; it must remain consistent with the probability path connecting the initial Gaussian distribution to the target action distribution (e.g., by satisfying the continuity equation). Consequently, learning a valid flow map implicitly requires learning a compatible marginal velocity field.

BFQ enforces this consistency

$$\pi(a_t, t - \Delta, t; s) \approx a_t - \Delta * v(a_t, t; s), \quad \text{as } \Delta \to 0 \tag{28}$$

However, directly learning displacement introduces a practical optimization issue. As $\Delta \to 0$, the displacement term $\Delta * v_\theta(\mathbf{x}_t, t, \mathbf{s})$ becomes vanishingly small, leading to weak gradients due to finite numerical precision. As a result, direct displacement-based optimization becomes unstable and difficult to train effectively.

To address this issue, we introduce a preconditioning mechanism that directly predicts the marginal velocity field $v_\theta$ without scaling by $\Delta$ (in the $\Delta \to 0$ regime). This preserves a strong learning signal while remaining consistent with the underlying probability path induced by Flow Matching. In practice, this design substantially improves optimization stability and learning effectiveness.

We empirically validate the importance of the proposed preconditioning mechanism through an ablation study in Table 6. Removing the preconditioning design leads to severe performance degradation across all evaluated datasets, demonstrating that the mechanism is crucial for stable and effective policy learning in practice.

*Table 6.* Effect of the proposed preconditioning mechanism in the policy architecture.

| Dataset | Preconditioning | No Preconditioning |
|---|---|---|
| HalfCheetah-Medium-Expert | $98.5 \pm 0.1$ | $3 \pm 3.0$ |
| HalfCheetah-Medium | $66.1 \pm 0.0$ | $14 \pm 0.3$ |
| HalfCheetah-Medium-Replay | $52.1 \pm 0.1$ | $16 \pm 0.1$ |

## G. Relation to Consistency Models, Shortcut Models, and Flow Map Matching

BQL is conceptually related to consistency models (Ding & Jin, 2023), shortcut models (Esser et al., 2024), and Flow Map Matching (FMM) (Boffi et al., 2024), as all aim to learn efficient long-range mappings across diffusion or flow trajectories. However, the underlying formulations and training objectives differ substantially.

Consistency models adopt a one-time-step formulation, where consecutive noisy states are encouraged to map to the same final action. In contrast, BFQ employs a two-time-step formulation $(t, r)$, which is more closely related to the formulations used in shortcut models and Flow Map Matching.

The shortcut model approximates displacement fields by discretizing time into finite intervals and learning long-range mappings through recursive midpoint composition (e.g., $0 \to d \to 2d$). Flow Map Matching instead enforces consistency implicitly through two coupled objectives: (1) velocity consistency after a round-trip (forward–backward), matching the true flow velocity (achieved through Jacobian-vector products), and (2) exact reconstruction of the starting point after the round-trip.

In contrast, BFQ enforces consistency explicitly by directly parameterizing and supervising the marginal velocity field. As $\Delta \to 0$, the learned objective approaches an explicit consistency condition over infinitesimal transitions while simultaneously supporting compositionality over arbitrarily long intervals. Importantly, BFQ achieves this behavior using only forward passes, avoiding the Jacobian-vector products required in FMM and the discrete composition procedures used in shortcut models.

## H. Implementation Details and Task-Specific Hyperparameters

We provide the implementation details for D4RL in Table 7.

| Hyperparameter | Value |
| --- | --- |
| Learning rate | 0.0003 |
| Optimizer | Adam |
| Gradient steps | 1,000,000 |
| Minibatch size | 256 |
| Policy and Q-function MLP dimensions | [256, 256, 256, 256] |
| Nonlinearity | Mish |
| Offset $\Delta_{\max}$ | 0.001 |
| $t, r$ embedding | Sinusoidal positional embeddings (64 dimensions) |
| Target network smoothing coefficient | 0.005 |
| Discount factor $\gamma$ | 0.99 (default), 0.995 (AntMaze-Giant, HumanoidMaze, AntSoccer) |
| Flow steps | 1 |
| Flow conditioning ratio $\lambda$ | 0.5 |
| Flow time sampling $(t, m, r)$ | Uniform([0, 1]) with $r \leq m \leq t$ |
| Clipped double Q-learning | False (default), True (D4RL locomotion, D4RL Adroit, D4RL AntMaze-{medium, large, giant}) |
| Normalization coefficient $\eta$ | Grid search over $\{0.001, 0.01, 0.05, 0.1, 0.3, 0.5, 1\}$ |

*Table 7.* Shared Hyperparameters used in our D4RL experiments.

We further provide all task-specific hyperparameters for OGBench in Table 8 and for D4RL in Table 9.

For OGBench and D4RL Adroit, we follow the experimental settings and reporting conventions of FQL (Park et al., 2025), adopting its recommended hyperparameters and baseline configurations. Briefly, $\alpha$ and $\beta$ control the strength of behavior cloning in the actor loss, $N$ denotes the number of candidates used in the Best-of-$N$ selection, and $\eta$ represents the scaling factor applied to the Q-value (i.e., $\alpha \cdot Q$). Further details on baselines and hyperparameter settings can be found in Appendix E.2 of FQL (Park et al., 2025) and SORL (Espinosa-Dice et al., 2026).

For D4RL Locomotion and AntMaze, the additional parameter $T$ denotes the number of denoising timesteps used in the diffusion process.

## I. Detailed Limitations and Future Work

The efficiency and expressivity of BFQ make it a promising framework for advancing practical reinforcement learning systems alongside existing approaches (Venkataraman et al., 2025; Nguyen et al., 2021; Luu et al., 2024; Nguyen et al., 2024b;a). In particular, BFQ enables one-step action generation while maintaining state-of-the-art performance, allowing decision-making at frequencies suitable for real-time applications such as robotics, autonomous driving, healthcare, and related domains (Ramstedt & Pal, 2019; Kiran et al., 2021; Ouyang et al., 2022). Moreover, its reduced training and inference cost lowers the computational requirements typically associated with generative reinforcement learning methods,

| Task | IQL ($\alpha$) | ReBRAC ($\alpha_1, \alpha_2$) | IDQL ($N$) | SRPO ($\beta$) | CAC ($\eta$) | FAWAC ($\alpha$) | FBRAC ($\alpha$) | IFQL ($N$) | FQL ($\alpha$) | SORL ($\alpha$) | BFQ ($\eta$) |
|---|---|---|---|---|---|---|---|---|---|---|---|
| antmaze-large-navigate-singletask-task1-v0 (*) | 10 | (0.003, 0.01) | 32 | 0.3 | 1 | 3 | 3 | 32 | 10 | 500 | 1 |
| antmaze-giant-navigate-singletask-task1-v0 (*) | 10 | (0.003, 0.01) | 32 | 0.3 | 1 | 3 | 10 | 32 | 10 | 500 | 1 |
| humanoidmaze-medium-navigate-singletask-task1-v0 (*) | 10 | (0.01, 0.01) | 32 | 0.3 | 0.03 | 3 | 30 | 32 | 30 | 100 | 0.001 |
| humanoidmaze-large-navigate-singletask-task1-v0 (*) | 10 | (0.01, 0.01) | 32 | 0.3 | 1 | 3 | 30 | 32 | 30 | 500 | 0.001 |
| cube-single-play-singletask-task2-v0 (*) | 1 | (1, 0) | 32 | 0.03 | 0.003 | 1 | 100 | 32 | 300 | 10 | 0.01 |
| scene-play-singletask-task2-v0 (*) | 10 | (0.1, 0.01) | 32 | 0.1 | 0.3 | 0.3 | 100 | 32 | 300 | 100 | 0.01 |
| pen-human-v1 | 3 | – | 32 | 0.03 | 0.003 | 0.03 | 30000 | 32 | 10000 | 10 | 0.001 |
| pen-cloned-v1 | 3 | – | 32 | 0.1 | 0.003 | 0.3 | 10000 | 32 | 10000 | 10 | 0.001 |

*Table 8.* Task-specific hyperparameters for OGBench and D4RL Adroit. We follow the settings and reporting conventions of FQL, adopting its recommended hyperparameters and baseline configurations. Briefly, $\alpha$ and $\beta$ control the strength of behavior cloning in the actor loss, $N$ denotes the number of candidates used in the Best-of-$N$ selection, and $\eta$ represents the scaling factor applied to the Q-value (i.e., $\alpha \cdot Q$). For further details on baselines and hyperparameters, please refer to Appendix E.2 of FQL and SORL.

| Task | TD3-BC ($\alpha$) | IQL ($\alpha$) | EDP ($\alpha, T$) | IDQL ($N$) | DQL ($\alpha$) | SRPO ($\beta$) | CAC ($\eta$) | SORL ($\alpha$) | FQL ($\alpha$) | BFQ ($\eta$) |
|---|---|---|---|---|---|---|---|---|---|---|
| HalfCheetah-Medium-Expert | 2.5 | 3 | (1, 15) | 50 | 1 | 0.01 | 1 | 100 | 0.1 | 0.05 |
| Hopper-Medium-Expert | 2.5 | 3 | (1, 15) | 50 | 1 | 0.01 | 1 | 150 | 1 | 0.01 |
| Walker2d-Medium-Expert | 2.5 | 3 | (1, 15) | 50 | 1 | 0.1 | 1 | 100 | 1 | 0.05 |
| HalfCheetah-Medium | 2.5 | 3 | (1, 15) | 50 | 1 | 0.2 | 0.1 | 100 | 0.03 | 1 |
| Hopper-Medium | 2.5 | 3 | (1, 15) | 50 | 1 | 0.05 | 1 | 100 | 0.3 | 0.05 |
| Walker2d-Medium | 2.5 | 3 | (1, 15) | 50 | 1 | 0.05 | 0.1 | 10 | 1 | 0.05 |
| HalfCheetah-Medium-Replay | 2.5 | 3 | (1, 15) | 50 | 1 | 0.2 | 1 | 100 | 0.03 | 1 |
| Hopper-Medium-Replay | 2.5 | 3 | (1, 15) | 50 | 1 | 0.2 | 0.1 | 50 | 1 | 0.05 |
| Walker2d-Medium-Replay | 2.5 | 3 | (1, 15) | 50 | 1 | 0.5 | 0.1 | 50 | 1 | 0.1 |
| AntMaze-Medium-Play | 5 | 10 | (2, 15) | 50 | 2 | 0.08 | 0.01 | 200 | 10 | 0.005 |
| AntMaze-Large-Play | 5.5 | 10 | (4.5, 15) | 50 | 4.5 | 0.06 | 4.5 | 400 | 3 | 0.3 |
| AntMaze-Medium-Diverse | 2.5 | 10 | (3, 15) | 50 | 3 | 0.05 | 0.01 | 300 | 10 | 0.001 |
| AntMaze-Large-Diverse | 3.5 | 10 | (3.5, 15) | 50 | 3.5 | 0.05 | 3.5 | 200 | 3 | 0.1 |

*Table 9.* Task-specific hyperparameters for D4RL Locomotion and AntMaze. The additional parameter $T$ denotes the number of denoising timesteps used in the diffusion process.

potentially facilitating deployment at larger scales and in more complex environments.

Despite these advantages, the current evaluation of BFQ is still primarily centered on single-goal, state-based control tasks using low-dimensional proprioceptive observations from the D4RL benchmark. Although we additionally provide experiments on higher-dimensional settings, including OGBench and D4RL Adroit, many important research directions remain open.

First, extending BFQ to online fine-tuning reinforcement learning is a natural next step. The proposed one-step formulation substantially reduces the computational overhead typically associated with generative policies, making BFQ particularly well-suited for real-time interaction and online data collection. However, the performance of BFQ—as with offline reinforcement learning methods more broadly—remains fundamentally constrained by dataset quality and coverage. Consequently, understanding optimization stability and sample efficiency in the online fine-tuning setting remains an important direction for future research.

Second, future work should explore applying BFQ to vision-based and point-cloud-based control problems involving high-dimensional observations. A key open question is whether BFQ can effectively leverage feature extractors from existing domain-specific architectures (He et al., 2016; Zhou & Tuzel, 2018; Vu et al., 2022; 2023; 2021), or whether new encoder designs specifically tailored for one-step flow-based policies are required. Addressing this challenge could enable scalable end-to-end learning in more complex and unstructured environments.

Third, extending BFQ to goal-conditioned and multi-task reinforcement learning represents another promising direction. In particular, learning conditional average velocity fields capable of supporting diverse goal-directed behaviors—without relying on separate diffusion models or auxiliary reward models—may provide improved flexibility and stronger generalization across tasks.

Overall, BFQ provides a general and computationally efficient foundation for expressive policy learning, and we hope future work further broadens its applicability across diverse reinforcement learning settings and potentially beyond reinforcement learning domains, including but not limited to visual understanding (Le et al., 2022; Kim et al., 2021; Yoon et al., 2022; Koo et al., 2024; Pham et al., 2022), audio processing (Latif et al., 2023; Sarı et al., 2021; Ton et al., 2025), and broader representation learning and multimodal modeling problems (Oquab et al., 2023; Zhang et al., 2022; Bui et al., 2026; Park

et al., 2026).

## J. Theoretical Discussion on Boundary Conditions

In this section, we provide additional theoretical intuition regarding the role of boundary conditions in BFQ.

**Boundary Conditions as Local Flow Anchors.** The proposed BFQ policy operator is trained through compositional consistency over arbitrary temporal intervals:

$$\pi(a_t, r, t; s) = \pi(\pi(a_t, m, t; s), r, m; s), \tag{29}$$

which admits many degenerate solutions if optimized alone. In particular, without additional constraints, the policy may satisfy compositional consistency while deviating from the underlying flow dynamics. Such solutions are undesirable because BFQ, although formulated as directly learning a flow map (i.e., a displacement operator), fundamentally models transport induced by an underlying marginal velocity field over time. Consequently, the learned transport cannot be arbitrary and must remain consistent with the probability path connecting the initial Gaussian distribution to the target action distribution. In particular, the corresponding velocity field must satisfy the continuity dynamics governing the transport process.

The boundary condition addresses this issue by anchoring the operator to the infinitesimal dynamics of the flow process:

$$\pi(a_t, t - \Delta, t; s) \approx a_t - \Delta * v(a_t, t; s), \quad \Delta \to 0. \tag{30}$$

This condition can be interpreted as a local first-order approximation of the transport dynamics. Consequently, the boundary supervision constrains the learned operator to remain locally consistent with the underlying continuous-time flow matching dynamics.

**Connection to Numerical Integration.** The proposed boundary condition is closely related to numerical integration of ordinary differential equations (ODEs). Recall that the underlying flow process satisfies:

$$\frac{da_t}{dt} = v(a_t, t; s). \tag{31}$$

For sufficiently small $\Delta$, Euler integration yields:

$$a_{t-\Delta} = a_t - \Delta * v(a_t, t; s) + \mathcal{O}(\Delta^2). \tag{32}$$

Therefore, the BFQ boundary objective effectively constrains the policy operator to behave as a first-order locally consistent integrator of the flow dynamics. This provides a principled initialization regime for learning larger transitions.

Importantly, while standard Flow Matching only learns local velocity fields, BFQ recursively composes short-range transitions to approximate long-range transport:

$$d(a_t, r, t; s) = \sum_{k=1}^{K} d(a_{t_k}, t_{k-1}, t_k; s), \tag{33}$$

where $r = t_0 < t_1 < \cdots < t_K = t$.

As a result, the boundary condition acts as the fundamental base case of the recursive decomposition.

**Future Directions.** Although the current work empirically demonstrates the effectiveness of boundary conditioning, a more formal analysis of convergence properties and approximation error propagation remains an important direction for future work. In particular, analyzing how local approximation errors accumulate through recursive composition may provide deeper theoretical insight into single-step flow-based policy learning.

## K. Full D4RL Results

We provide the complete D4RL results, including additional comparisons with CAC (Ding & Jin, 2023), for reference in Table 10.

| Policy Type | Gaussian Policy | | | Diffusion Policy | | | | One-Step Flow/Diffusion Policy | | | | | |
|---|---|---|---|---|---|---|---|---|---|---|---|---|---|
| Dataset | BC | TD3-BC | IQL | EDP | IDQL | DQL | CAC-2 | CAC | SRPO | SORL | OFQL | FQL | BFQ (Ours) |
| HalfCheetah-Medium-Expert | 60.2±13.2 | 91.5±15.8 | 88.3±2.8 | 95.8±0.1 | 91.3±0.6 | 95.5±0.1 | 89.2±3.3 | 47.9±10.6 | 92.2±3.0 | 96.5±0.9 | 95.2±0.4 | **99.8±0.1** | 98.6±0.1 |
| Hopper-Medium-Expert | 67.2±20.6 | 101.6±23.2 | 76.6±34.9 | 110.8±0.4 | 110.1±0.7 | **111.1±0.4** | 106.0±1.3 | 0.1±0.0 | 100.1±13.9 | 45.9±6.7 | 110.2±1.3 | 86.2±1.3 | 110.5±1.5 |
| Walker2d-Medium-Expert | 105.4±3.1 | 110.4±0.4 | 108.7±2.2 | 110.4±0.0 | 110.6±0.0 | 111.6±0.9 | 111.6±0.7 | -1.3±0.3 | 114.0±2.1 | 109.1±1.7 | 113.0±0.1 | 100.5±0.1 | **113.4±0.1** |
| HalfCheetah-Medium | 42.5±0.2 | 48.5±0.7 | 47.7±0.2 | 50.8±0.0 | 51.5±0.1 | 52.3±0.2 | **71.9±0.8** | 62.2±0.7 | 60.4±0.8 | 57.4±0.7 | 63.8±0.1 | 60.1±0.1 | 66.1±0.0 |
| Hopper-Medium | 52.9±0.1 | 56.6±9.0 | 61.2±6.4 | 72.6±0.2 | 70.1±2.0 | 96.5±1.3 | 99.7±2.3 | 46.3±31.6 | 95.5±2.0 | 81.3±5.8 | **103.6±0.1** | 74.5±0.2 | 103.5±0.1 |
| Walker2d-Medium | 72.3±3.2 | 83.3±7.0 | 78.7±4.5 | 86.5±0.2 | 86.1±0.4 | 86.8±0.0 | 84.1±0.3 | 86.6±7.2 | 84.4±4.4 | 77.9±2.5 | 87.4±0.1 | 72.7±0.8 | **91.7±0.1** |
| HalfCheetah-Medium-Replay | 34.7±1.8 | 44.5±0.8 | 42.9±1.7 | 44.9±0.4 | 46.5±0.3 | 47.9±0.0 | **62.7±0.6** | 47.9±0.6 | 51.4±3.4 | 48.3±0.2 | 51.2±0.1 | 51.1±0.1 | 52.1±0.1 |
| Hopper-Medium-Replay | 19.7±5.5 | 55.2±24.6 | 86.8±15.5 | 83.0±1.7 | 99.4±0.1 | 91.2±0.0 | 100.4±0.6 | 91.7±5.1 | 101.2±1.0 | 93.0±2.1 | 101.9±0.7 | 85.4±0.5 | **102.1±0.1** |
| Walker2d-Medium-Replay | 29.0±0.5 | 82.5±13.6 | 68.3±6.4 | 87.0±2.6 | 89.1±2.4 | 98.2±0.1 | 83.0±1.5 | 86.6±7.2 | 84.6±7.1 | 67.8±11.8 | **106.2±0.6** | 82.1±1.2 | 97.5±1.2 |
| **Average (MuJoCo)** | 53.8 | 74.9 | 73.2 | 82.4 | 84.1 | 89.0 | 89.8 | 52.0 | 87.1 | 80.0 | 92.5 | 79.2 | **92.8** |
| AntMaze-Medium-Play | 0.0±0.0 | 10.6±10.1 | 75.5±5.5 | 73.3±6.2 | 67.3±5.7 | 86.0±1.8 | 49±24 | 0 | 80.7±7.1 | 80.1±11.6 | **88.1±5.0** | 78.0±7.0 | 87.0±4.0 |
| AntMaze-Large-Play | 0.0±0.0 | 0.2±0.4 | 38.6±4.2 | 33.3±1.9 | 48.7±4.7 | 83.3±2.5 | 0±1 | 0 | 53.6±12.5 | 57.3±16.4 | 84.0±6.1 | 84.0±7.0 | **88.5±4.8** |
| AntMaze-Medium-Diverse | 0.6±0.2 | 5.7±8.2 | 78.0±1.5 | 52.7±1.9 | 83.3±5.0 | 94.7±2.5 | 0±0 | 0 | 75.0±12.3 | 70.0±8.1 | **90.2±4.2** | 71.0±13.0 | 84.1±6.9 |
| AntMaze-Large-Diverse | 0.0±0.0 | 0.0±0.0 | 48.0±7.5 | 41.3±3.4 | 40.0±11.4 | 61.3±8.4 | 0 | 0 | 53.6±6.3 | 52.5±10.9 | 76.1±6.6 | **83.0±4.0** | 76.1±7.0 |
| **Average (AntMaze)** | 0.2 | 4.1 | 60.0 | 50.2 | 59.8 | 81.3 | 12.3 | 0 | 73.6 | 65.0 | **84.6** | 79.0 | 83.9 |

*Table 10.* Normalized performance on the D4RL benchmark across MuJoCo and AntMaze domains. Bold values indicate the best result per task. CAC-2 denotes the two-step inference variant of CAC.

