# OpenReview forum: "Fast and Highly Expressive Policy Learning for Offline Reinforcement Learning via Bootstrapped Flow Q-Learning"
_ICML.cc/2026/Conference — ICML 2026 regular_

### Official Review · Reviewer_3DUb · 2026-03-03

**Soundness:** 2
**Presentation:** 2
**Significance:** 3
**Originality:** 2
**Overall Recommendation:** 4
**Confidence:** 3

**Summary:**

In this paper, author purposes a bootstrapped Flow matching Q-learning framework, which eliminates the need for multistep denoising, making the learning process substantially faster, simpler, and more robust.

**Compliance With Llm Reviewing Policy:**

Affirmed.

**Final Justification:**

Authors' rebuttal and additional experiments have solved majority of my questions and concerns. I think the paper is ready if the presentation is fixed as authors have promised.

**Key Questions For Authors:**

See below in the limitation section.

**Limitations:**

1, More algorithmic details are needed, such as the the overall training framework, etc. \
2, Section 4 needs some rewriting, since current three paragraphs on policy modeling, Compositional Consistency, and Boundary Conditions all seem detached from each other. It would be better to have a heuristic overview on the connection in between.\
3, It would be interesting to explore on the theoretical side of the boundary conditions, particularly boundary conditions effectiveness on the policy operator learning. \
4, For the empirical studies, the investigation on pure D4RL mujoco locomotion suite is bit trivial, it would be better to extend to Adroit hand or other more complex tasks.

**Strengths And Weaknesses:**

1, The Methodology illustration is straightforward and intuitive, but the overall writing needs some improvement. \
2, The paper structure is pretty completed with great logic coherence. \
3, The empirical study provides comprehensive comparison with various methods/approaches, and the bandit action example is straightforward.

---

> ### Author Rebuttal · Authors · 2026-03-31
>
> Thank you for the detailed review and constructive feedback on this work.
>
> **Q1.** More algorithmic details are needed, such as the overall training framework, etc.
>
> Our method is based on a behavior-regularized actor-critic framework, described in Section 3. We also provide full pseudocode in Appendix D. To address this concern, we will add more details and better guide readers to the relevant sections.
>
> **Q2.** Section 4 needs clearer connections between components.
>
> We agree that the current presentation can be improved. In the revision, we will provide a clearer high-level overview to explicitly connect policy modeling, compositional consistency, and boundary conditions, and better illustrate how they jointly define our formulation.
>
> **Q3.** It would be interesting to explore on the theoretical side of the boundary conditions, particularly boundary conditions effectiveness on the policy operator learning.
>
> We agree that further theoretical analysis of boundary conditions is valuable and will be included in the revision.
>
> Intuitively, Eq. (16) enforces compositional consistency, enabling learning from short transitions and generalization to longer horizons. Eq. (17) defines the boundary condition at small time intervals, ensuring consistency with the underlying flow dynamics. Eq. (18) provides a practical realization via conditional velocity, yielding a stable training signal.
>
> Together, these constraints regularize the learned policy operator and guide it toward consistent and well-behaved dynamics. We will further clarify this theoretical perspective in the revised manuscript.
>
>
> **Q4.** Extending empirical studies beyond D4RL MuJoCo locomotion.
>
> We conduct experiments on OGBench, selecting six representative tasks spanning diverse domains: navigation (AntMaze-Large, AntMaze-Giant), high-dimensional locomotion (HumanoidMaze-Medium, HumanoidMaze-Large), and robot manipulation (Cube-Single, Scene-Play). We further evaluate our method on the D4RL Adroit benchmark (pen-human, pen-cloned). All results are reported in Table 2:
> https://anonymous.4open.science/r/AnonymousICML2026-FFFFFF/MainResultAndAblation.pdf
>
> On OGBench, BFQ achieves strong performance across all task categories despite using a single-step policy. It attains the highest average score (68), outperforming strong baselines such as SORL-8 (63) and FQL (47). BFQ also achieves top or near-top results on most tasks (e.g., 94 on humanoidmaze-medium and 91 on antmaze-large), demonstrating that it can match or surpass multi-step methods while offering one-step inference.
>
> On the D4RL Adroit benchmark, BFQ shows strong and consistent performance across both tasks, achieving the highest average score (79). It obtains the best result on pen-human-v1 (82 ± 7), surpassing FBRAC (77 ± 7) and IDQL (76 ± 10), and remains competitive on pen-cloned-v1 (75 ± 7), close to IFQL (80 ± 11) while maintaining a simpler one-step policy.

---

> > ### Author Rebuttal · Reviewer_3DUb · 2026-04-02
> >
> > Authors have added thorough add-up experiments and presentation revisions, which solved majority of my questions and concerns. I have updated my score accordingly.

---

### Official Review · Reviewer_Nfyu · 2026-03-10

**Soundness:** 2
**Presentation:** 3
**Significance:** 3
**Originality:** 2
**Overall Recommendation:** 3
**Confidence:** 4

**Summary:**

This paper introduces BFQ. BFQ learns a noise-to-action displacement  via compositional bootstrapping,  Dueing to its single step inference, DFQ bypass BPTT . Results on part of D4RL tasks are competitive.

**Compliance With Llm Reviewing Policy:**

Affirmed.

**Final Justification:**

Thank you to the authors for their rebuttal. I believe some of the issues I raised were not adequately addressed in either the original paper or the rebuttal, so I stand by my score.

**Key Questions For Authors:**

Why not just use original consistency models/shortcut models(the claim of  without of distillation procedures is not convicing ,you actually use the similar self-distillation methods like shortcut models/consistency models)? What is  the unique benefits of your formulation? Can you provide a clear ablation or comparsion to previous one step method?

For others, see weakness

**Limitations:**

yes

**Strengths And Weaknesses:**

## Strengths

-  Practical efficiency gains.  Single-step inference  and BPTT-free training  are meaningful for deployment-constrained settings. The memory savings from avoiding backpropagation through the sampling chain are a tangible engineering benefit.

- Reasonable empirical performance.  Results on MuJoCo and AntMaze are competitive with existing multi-step diffusion baselines. The ablation study (Table 2) is adequately designed, confirming that each proposed component (compositional consistency, boundary loss, preconditioning) contributes to the final performance.

-  Clear presentation. The three-panel Figure 2 provides a useful overview of the method. The paper is generally well-organized and easy to follow.

## Weaknesses

**W1. The formulation follows the idea of flow map, and the method's connections（relationship and difference） to prior flow map learning and generative modeling literature are not discussed.**
Its three training objectives correspond to well-established flow map/consistency model/shortcut models properties.
This similarity is ok, but the authors should discuss the relationships and differences between them carefully.

A further question arises from the implementation: if the goal is to predict a displacement, why does the preconditioning (Eq. 20) re-parameterize the network to output a velocity field ? The Flow Map Matching framework does not require such a trick to handle boundary condition.This trick reduces the method back to a setting closer to Consistency Models  and Shortcut Models(despite differences in training details). The paper does not discuss any of these relationships, nor distinguish its contribution from these prior methods, risking the impression that these are novel ideas rather than known techniques applied to offline RL.

**W2. Performance attribution is incomplete, the source of  gains  over methods sharing the same offline rl paradigm is not analyzed.**

BFQ, FQL, and DQL all follow a similar TD3+BC-style actor-critic paradigm, where the generative policy's advantage comes from capturing multimodal distributions in the dataset. FQL further reduces the optimization difficulty caused by BPTT via one step distillation. These methods are theoretically within the same paradigm, yet BFQ shows disproportionately large improvements on certain tasks. Is this because the one-step policy learned via compositional bootstrapping is more expressive than FQL's distilled policy? Or is it purely an optimization stability effect from eliminating BPTT entirely? A deeper discussion of why and when BFQ outperforms other single-step methods within this shared paradigm would strengthen the contribution.

**W3. Baselines lack single-step generative RL methods.**

All baselines except fql are multi-step.  Methods that use meanflows or shortcut models(for example:SORL OFQL)  for offline rl is misssing. Additionally, D4RL locomotion and AntMaze are increasingly saturated. Recent generative-policy work  has adopted OGBench. Evaluation on OGBench would be more informative.

---

> ### Author Rebuttal · Authors · 2026-03-31
>
> Thank you for the detailed review and constructive feedback on this work.
>
> **W1.** While inspired by consistency and shortcut models, BFQ differs in key aspects.
>
> Consistency models (one-step) are often unstable, especially in RL. BFQ instead uses a two-time-step formulation, improving stability while retaining efficient inference.
> Compared to shortcut models, BFQ enforces consistency over continuous triplets (t,m,r) rather than a fixed midpoint, leading to a more flexible formulation and better performance.
> Finally, unlike Mean Flow methods requiring Jacobians, BFQ uses only forward passes, resulting in a simpler and more stable training pipeline.
>
> **W2.** Empirically, DQL requires multiple denoising steps (e.g., 5) and relies on backpropagation through time, resulting in slow training and suboptimal performance, as also observed in the FQL paper. Moreover, DQL cannot operate in a one-step setting.
>
> Regarding FQL, we argue that a one-step policy learned via compositional bootstrapping is more expressive than FQL’s distilled policy. Empirically, our method outperforms FQL on both D4RL and OGBench (see Tables 1–3):
> https://anonymous.4open.science/r/AnonymousICML2026-FFFFFF/MainResultAndAblation.pdf
>
> To further verify this, we conduct experiments on a toy dataset:
> https://anonymous.4open.science/r/AnonymousICML2026-FFFFFF/Ablation_on_FQLvsBQL.pdf
>
> The results show that the one-step FQL policy fails to capture complex checkerboard patterns. Although the 10-step flow matching model captures the distribution well, the one-step model struggles to distill it, leading to compounded errors (flow matching error + distillation error). In contrast, the one-step BQL policy successfully captures the complex distribution.
>
>
> **Q1.** Why not just use original consistency models/shortcut models?
>
> Consistency models are known to be unstable in one-step training, especially in RL (e.g., Consistency AC struggles in this regime).
> While BFQ shares similarities with consistency/shortcut models, it is empirically more stable and performs better. In particular, unlike shortcut models that enforce consistency at a fixed midpoint, BFQ enforces consistency over continuous triplets  (t,m,r), leading to a more flexible formulation.
> Overall, BFQ adopts a more direct and stable training formulation, which improves performance in practice.
>
> **Q2.** What are the unique benefits of the proposed formulation?
>
> Please refer to our responses to **W1** and **W2** for details
>
> **Q3.** Can you provide a clear ablation or comparison to previous one-step methods?
>
> We include comparisons with various one-step methods; please refer to the following supplementary material:
> https://anonymous.4open.science/r/AnonymousICML2026-FFFFFF/MainResultAndAblation.pdf

---

> > ### Author Rebuttal · Reviewer_Nfyu · 2026-04-02
> >
> > Thank you for the rebuttal. The additional results are helpful, but my main concerns remain.
> > - I still do not understand why a velocity-field parameterization is needed if the method is conceptually framed as learning a direct displacement / flow map. This is important because it affects how different BFQ really is from prior consistency/shortcut/flowmap-style methods.
> > - The stability claim is still not convincing. The rebuttal says BFQ is more stable, but does not provide a reasonable explanation. If shortcut/consistency-style methods can be trained in large-scale generative modeling, why are they unsuitable here even in smaller networks? Is the issue actor-critic coupling, critic non-stationarity, boundary behavior, or hyperparameter sensitivity? This needs clearer analysis.
> > - The continuous-triplet formulation currently looks like a relatively simple modification of shortcut-style midpoint decomposition. I really love simple but effective ideas, but then the paper should explain what property this modification changes and why it improves stability or performance.
> > - The relation to prior flow-map / consistency / shortcut literature should be discussed explicitly. This is necessary for proper scholarly positioning.
> > - Finally, code and sufficiently detailed hyperparameter settings are necessary for proper scientific evaluation and reproducibility. Without them, I cannot judge whether the empirical comparisons are fully fair, especially in offline RL, where results are often highly sensitive to implementation and tuning details.
> > Overall, I appreciate the clarifications, but the rebuttal does not fully resolve my concerns about mechanism, positioning with respect to prior work, and performance attribution.
> > Since I still need to check six papers, if I have overlooked or misunderstood anything, please let me know. I will do my best to reply within 12 hours.

---

> > > ### Author Response · Authors · 2026-04-02
> > >
> > > We understand your busy schedule and are grateful for your clarification and active support of the community. We wish you the best with your rebuttals. Below, we provide elaborations:
> > >
> > > 1. Why is velocity-field parameterization needed?
> > >
> > > While BFQ is conceptually framed as learning a direct flow map (i.e., displacement), this displacement is the time integral of an underlying marginal velocity field. Importantly, the velocity field cannot be arbitrary—it must remain **consistent with the probability path** (e.g., satisfy the continuity equation) that connects the initial Gaussian to the target action distribution. Hence, learning a valid flow map implicitly requires respecting the marginal velocity.
> > >
> > > BFQ enforces this via:
> > >
> > > $$\pi(a_t, t - \Delta, t; s) \approx a_t - \Delta * v(a_t, t; s), \quad \text{as } \Delta \to 0.$$
> > >
> > > However, as Δ→0, the term Δ* $v(a_t, t; s)$ becomes vanishingly small, leading to weak gradients due to finite numerical precision. This makes direct displacement-based learning unstable.
> > >
> > > Our architectural preconditioning helps resolve this issue in practice, where the model predicts $v(a_t, t; s)$ and supervises it directly without scaling by Δ (only in the Δ→0 limit), preserving a strong learning signal while remaining consistent with the underlying probability path.
> > > Empirically, we validate this via ablation:
> > > |Dataset|Preconditioning|No Preconditioning|
> > > |---|---|---|
> > > |HC-ME|98.5 |3|
> > > |HC-M|66.1 |14|
> > > |HC-MR|52.1 |16|
> > >
> > > Removing preconditioning leads to a significant performance drop, confirming that it is critical for stable learning.
> > >
> > > 2. On the stability claims.
> > >
> > > We focus on the offline RL setting and base our claims on empirical evidence from D4RL (Table 1), OGBench (Table 2), and hyperparameter analysis (Table 3):
> > >
> > > https://anonymous.4open.science/r/AnonymousICML2026-FFFFFF/MainResultAndAblation.pdf
> > >
> > > These conclusions may not directly transfer to supervised domains such as image generation (or vice versa), which differ substantially from our setting.
> > >
> > > Empirically, consistency methods struggle in RL, especially in one-step settings (see Table 1). For example, CAC (i.e., Consistency AC) yields near-zero or negative returns on tasks such as Hopper-Medium-Expert (0.1) and Walker2d-Medium-Expert (-1.3), and performs poorly on AntMaze (0.0), even with two-step denoising CAC-2 (12.3). In contrast, BFQ consistently achieves much higher scores, supporting the stability claim over CAC and CAC-2. Similar trends are observed for shortcut models: SORL (1-step) underperforms BFQ across D4RL locomotion (80.0 vs. 92.8), navigation (65.0 vs. 83.9), and OGBench (43 vs. 68), and still lags behind even with 8 steps OGBench (SORL-8: 63 vs. BFQ: 68). Since these methods share the AC framework, the results are likely influenced by how expressive behavior cloning is formulated. We argue that BFQ enforces consistency over continuous triplets (t,m,r), rather than a fixed midpoint as in shortcut models, modeling finer-grained intervals and improved performance.
> > >
> > > Furthermore, regarding hyperparameter-related stability, we provide an ablation over η on HalfCheetah (see Table 3). The results show that BFQ is relatively easy to tune, with a wide range of η values yielding strong performance.
> > >
> > > 3.  Property of continuous-triplet formulation to shortcut-style midpoint decomposition.
> > >
> > > The model is trained by sampling arbitrary continuous triplets (r,m,t) with 0≤r≤m≤t≤1, and enforcing consistency across these intervals. As a result, BFQ implicitly learns relationships over infinitely many possible intervals, rather than a fixed discrete set (Shortcut Model). Empirically, we observed better performance (see response 2), and a hyperparameter (num discrete intervals) is not needed.
> > >
> > > 4. The relation to flow-map/consistency/shortcut model.
> > >
> > > Consistency models follow a one-time-step formulation, enforcing consecutive noisy states $z_t; z_{t+1}$ to map to the same final action. In contrast, BFQL adopts a two-time-step formulation (t,r), more aligned with Flow Map Matching and Shortcut Model. The shortcut model approximates displacement by discretizing time into finite intervals and learning long-range mappings via a fixed midpoint composition (0→d→2d). Flow Map Matching (FMM) enforces consistency implicitly via: (1) velocity consistency after a round-trip (forward–backward), matching the true flow velocity (achieved through Jacobian-vector products), and (2) exact reconstruction of the starting point after the round-trip. In contrast, as Δ→0, BFQL enforces consistency explicitly by directly parameterizing and supervising the marginal velocity field, combined with infinite-interval compositionality over longer ranges, using only forward passes.
> > >
> > > 5. Code/Detailed hyperparameter settings.
> > >
> > > We provide all task-specific hyperparameters for OGBench and D4RL; please refer to Tables 5, 6, and 7.
> > >
> > > https://anonymous.4open.science/r/AnonymousICML2026-FFFFFF/Hyperparameters.pdf
> > >
> > > The code will be released upon acceptance.

---

### Official Review · Reviewer_d95U · 2026-03-10

**Soundness:** 3
**Presentation:** 3
**Significance:** 3
**Originality:** 3
**Overall Recommendation:** 5
**Confidence:** 5

**Summary:**

The paper tackles a limitation of standard flow-based policy learning, where the learned marginal velocity field often induces curved generation trajectories. To address this issue, the authors propose learning the one-step displacement from noise to action through the composition of short-range displacements, while employing dynamic velocity matching to satisfy boundary conditions. Experimental results on D4RL demonstrate the effectiveness of the approach, and inference-time comparisons with prior methods further suggest its computational efficiency.

**Compliance With Llm Reviewing Policy:**

Affirmed.

**Final Justification:**

The rebuttal has addressed most of my concerns. I decided to raise my score to 5 and keep a high confidence.

**Key Questions For Authors:**

1. Is the proposed policy architecture necessary? What are its advantages over simply pretraining a velocity-field network?

2. I can understand why the proposed method may have a lower decision frequency than 1-step FQL due to its more complex policy architecture. However, why is its training time still comparable to that of 5-step FQL?

**Strengths And Weaknesses:**

**Strengths:**
1. The paper is well motivated and clearly written, and it addresses an important problem in the field.
2. The proposed policy learning approach, which composes short-range displacements, is simple yet effective, and is well supported from both intuitive and mathematical perspectives. The empirical results are also promising.
3. The proposed method achieves more efficient inference.

**Weaknesses:**
1. The experimental evaluation lacks results on more challenging benchmarks, such as Adroit and OGBench.
2. The paper does not provide experiments of the hyperparameter $\eta$ and the value choices for different tasks.

---

> ### Author Rebuttal · Authors · 2026-03-31
>
> Thank you for the detailed review and constructive feedback on this work.
>
> **Q1.**  Is the proposed policy architecture necessary? What are its advantages over simply pretraining a velocity-field network?
>
> The proposed architecture is not strictly necessary from a theoretical standpoint, but it is critical for practical and stable optimization. Empirically, removing this design causes severe performance drops, confirming its importance.
>
> | Dataset                    | Preconditioning | No Preconditioning |
> |---------------------------|----------------|--------------------|
> | HalfCheetah-Medium-Expert | 98.5 ± 0.1     | 3 ± 3.0   |
> | HalfCheetah-Medium        | 66.1 ± 0.0     | 14 ± 0.3    |
> | HalfCheetah-Medium-Replay | 52.1 ± 0.1     | 16 ± 0.1   |
>
> Regarding BFQ advantages over pretraining a velocity-field network: Pretraining a separate velocity-field model is feasible but introduces additional complexity (an extra training phase and model) and computational overhead. In contrast, BFQ uses conditional Flow Matching velocity as built-in supervision, enabling a single-stage, single-model framework that jointly learns velocity and displacement. This design is simpler, more efficient, and empirically effective.
>
> **Q2.** Why is BFQ training time still comparable to that of 5-step FQL?
>
> The target label in BFQ is computed via **calling the model**, which introduces some additional overhead beyond the standard forward pass during training. Crucially, this overhead is **fixed**—it does **not scale** with the number of diffusion or denoising steps, unlike FQL or DQL. In complex tasks, these methods may require **up to 100 denoising steps**, leading to significantly longer training times, whereas BFQ avoids this scaling issue and thus remains comparably efficient.
>
> **W1.** We provide experiments on OGBench and Adroit; please refer to Table 2
>
> https://anonymous.4open.science/r/AnonymousICML2026-FFFFFF/MainResultAndAblation.pdf
>
> On OGBench, BFQ achieves strong performance across all task categories despite using a single-step policy. It attains the highest average score (68), outperforming strong baselines such as SORL-8 (63) and FQL (47). BFQ also achieves top or near-top results on most tasks (e.g., 94 on humanoidmaze-medium and 91 on antmaze-large), demonstrating that it can match or surpass multi-step methods while maintaining one-step inference.
>
> On the D4RL Adroit benchmark, BFQ shows consistent and strong performance, achieving the highest average score (79). It obtains the best result on pen-human-v1 (82 ± 7), surpassing FBRAC (77 ± 7) and IDQL (76 ± 10), and remains competitive on pen-cloned-v1 (75 ± 7), close to IFQL (80 ± 11) while retaining a simpler one-step policy.
>
> **W2.** We provide all task-specific hyperparameters for OGBench and D4RL; please refer to Tables 5, 6, and 7.
>
> https://anonymous.4open.science/r/AnonymousICML2026-FFFFFF/Hyperparameters.pdf
>
> **Hyperparameter Study.**
> As noted in Line 377, the main hyperparameters are the boundary conditioning ratio λ and coefficient η. We already provide an ablation for λ in *On Boundary Conditioning Ratio*.
>
> For η, it typically depends on dataset suboptimality, consistent with prior offline RL practice. We therefore include an ablation over η on HalfCheetah:
>
> | Dataset                    | 0.01        | 0.05        | 0.1         | 0.5         | 1            | 5            | 10           |
> |----------------------------|------------|------------|------------|------------|--------------|--------------|--------------|
> | HalfCheetah-Medium-Expert | **95.1 ± 0.2** | **98.5 ± 0.1** | **95.6 ± 0.1** | 69.7 ± 3.5 | 68.2 ± 1.8  | 56.5 ± 4.2  | 47.7 ± 3.5  |
> | HalfCheetah-Medium        | 50.1 ± 0.05 | 58.1 ± 0.08 | **60.1 ± 0.1** | **65.1 ± 0.1** | **66.1 ± 0.0** | **63.6 ± 0.2** | 59.1 ± 2.1 |
> | HalfCheetah-Medium-Replay | 44.3 ± 0.1  | 46.7 ± 0.1  | **48.5 ± 0.1** | **51.1 ± 0.1** | **52.1 ± 0.1** | **51.1 ± 0.2** | **49.1 ± 1.1** |
>
> BFQ is relatively easy to tune, with a wide range of η yielding strong performance. Larger η works better for more suboptimal datasets (Medium, Medium-Replay), while smaller η is preferred for higher-quality data (Medium-Expert).

---

> > ### Author Rebuttal · Reviewer_d95U · 2026-04-01
> >
> > My main concern is the performance of other challenge tasks, and the author provided detailed results. So I raise my score and support that this paper is qualified to be accepted.

---

### Official Review · Reviewer_er13 · 2026-03-12

**Soundness:** 3
**Presentation:** 1
**Significance:** 3
**Originality:** 3
**Overall Recommendation:** 4
**Confidence:** 4

**Summary:**

This paper proposes an interesting idea to boostrap from a marginal velocity field to obtain  an one-step generation policy.

**Compliance With Llm Reviewing Policy:**

Affirmed.

**Final Justification:**

The rebuttal has addressed most of my concerns, and I have increased to 4. I hope authors could make sure to imporve the writing for the revised manuscript.

**Key Questions For Authors:**

Questions:
1. How does this compare to meanflow policy?
2. How many intervals do you need to for boostrapping?
3. "To normalize across dataset-specific Q-value scales, the
coefficient α is further adapted as:" Is this term proposed by yourself or from prior work? If so, please cite it.
4. In Table 1, some results have standard deviations, some do not? Why?

**Limitations:**

yes

**Strengths And Weaknesses:**

Strength:
1. Based on my knowledge, this idea is interesting and novel!

Weakness:

1. Writing needs to be improved. It takes me much amount of time to understand your equations. I have a few suggestions:


   1.1, notations for interval  ∆, seems to be confusing. for example  ∆ v(at, t; s), it is hard to tell whether this is small change in velcoity or delta*vlecoity. Could you change it?


   1.2 I think the idea is intuitive but the way of explaning it is not.Especially, for equations (16), (17), (18), very minor efforts are given to explain these. please consider explain them in much more details, I think it is very important for understanding.


   1.3 Figure 2 (2) seems messy and hard to understand its meaning. Figure (1) should also gives more information, in its current form seems does not help to understand the approach.


   1.4 The disccusions on Boundary condition via conditional velocity is a liitle bit confusing. Could you please add more motivations for this design? If you first learn a flow velcoity field as nomrally done, then you could still utilize it as the boundary condition, is this right? But uses conditional velocity seems also fine, as long as you mix them.

2. In Equation (19), what is the distribution of lambda? equation (19) should be in expectation? why do you design lambda to be random at all?

3. "We adopt an MLP-based
architecture for policy modeling. Specifically, the policy
is parameterized as a 4-layer multilayer perceptron (MLP)
with Mish activations and 256 hidden units per layer. The
policy input is formed by concatenating the action latent
vector, the current state vector, and sinusoidal positional embeddings of timesteps t and r (with embedding dimension
64). The policy outputs either a latent action or a velocity,
depending on whether the model is queried to predict the
action at timestep t or to anchor to marginal velocity. We
employ two Q-networks, each implemented as a 4-layer
MLP with Mish activations and 256 hidden units per layer.
Each Q-network takes as input the concatenation of the action and observation vectors and outputs a scalar Q-value
estimate. For timestep sampling, t, r, and m are drawn from
a uniform distribution subject to 0 ≤ r < m < t ≤ 1. The
offset ∆ for boundary condition is drawn from [0, ∆max],
and we empirically observe that ∆max = 1e − 3 provides
robust performance across all environments." Maby have a table for all hyperparameters to make it clear?

4. If i understand it correctly, the results come from 3 random seeds. Please consider addign more seeds.

---

> ### Author Rebuttal · Authors · 2026-03-31
>
> Thank you for the detailed review and constructive feedback on this work.
>
> **Q1.** How does this compare to the MeanFlow policy?
>
> Our approach shares some similarities with recent high-performing two-time-step MeanFlow methods such as OFQL [1]. Empirically, BFQ achieves slightly better performance on D4RL locomotion tasks (92.7 vs. 92.5 on average) and remains comparable on AntMaze navigation (83.9 vs. 84.6) compared to OFQL. Full results are provided in our response to Q4.
>
> However, the key differences lie in the underlying formulation and practicality.
>
> MeanFlow learns an average velocity field, which relies on Jacobian-based computations (e.g., JVPs). In contrast, BFQ directly models state transitions, requiring only standard forward passes and avoiding Jacobians or higher-order derivatives.
>
> This distinction is important in practice. Jacobian-based operations introduce additional computational and implementation overhead, and can be less compatible with modern training techniques (e.g., Flash Attention, Gradient checkpointing) [2], especially when scaling to large foundation encoders. Moreover, MeanFlow typically requires careful design choices—such as adaptive loss weighting and specialized time sampling (e.g., logit-normal)—which increase tuning complexity.
>
> [1] One-Step Flow Q-Learning: Addressing the Diffusion Policy Bottleneck in Offline Reinforcement Learning
>
> [2] https://github.com/Dao-AILab/flash-attention/issues/1672
>
> **Q2.** How many intervals do you need for bootstrapping?
> BFQ does not require explicit discretization of the time domain into a fixed number of intervals. The model is trained by sampling arbitrary continuous triplets (r,m,t) with 0≤r≤m≤t≤1, and enforcing consistency across these intervals. As a result, BFQ implicitly learns relationships over infinitely many possible intervals, rather than a fixed discrete set.
>
> **Q3.** This normalization strategy is not newly proposed in our work, but is adopted from prior literature, specifically **TD3-BC** and  **DQL**. We have already cited these works and will revise the manuscript to make this connection more explicit.
>
> **Q4.** We follow prior works (DQL, SRPO, OFQL, CAC, IQL) and do not report variance for non-diffusion baselines, as they are typically presented without standard deviations and consistently underperform diffusion-based methods.
>
> To address the concern, we provide additional runs with standard deviations here:
> https://anonymous.4open.science/r/AnonymousICML2026-FFFFFF/MainResultAndAblation.pdf
>
> **W1.1.** The notation denotes Δ⋅× velocity, we will revise the notation from Δv(at​,t;s) to Δ* v(at,t;s)
>
> **W1.2.**  We will revise the manuscript to better connect Eqs. (16)–(18)  to compositional consistency and boundary conditions via conditional velocity.
>
> Specifically, Eq. (16) enforces compositional consistency: a transition from t to 𝑟 should equal the composition of transitions 𝑡→𝑚→𝑟, enabling learning from short steps and generalization to longer horizons.
> Eq. (17) defines the boundary condition for small time intervals, ensuring the policy reduces to a local flow step consistent with flow matching dynamics.
> Finally, Eq. (18) replaces this ideal constraint with a practical velocity-matching objective using conditional Flow Matching velocity, providing a stable training signal without requiring a pretrained teacher.
>
> **W1.3.** We revise the two Figures accordingly
>
> https://anonymous.4open.science/r/AnonymousICML2026-FFFFFF/Figure_1_Revision.pdf
>
> https://anonymous.4open.science/r/AnonymousICML2026-FFFFFF/Figure_2_Revision_.pdf
>
> **W1.4.** The key motivation is that BFQ learns displacement, which corresponds to the integral of a velocity field that must remain consistent with the underlying probability path. While one could pretrain a separate flow model and use its velocity as a boundary condition, this introduces an additional model. Instead, we use conditional velocity from Flow Matching as a built-in supervision signal, enabling a single-stage, single-model training pipeline that is simpler and more efficient.
>
> **W2.** $\lambda$ is **not a random variable**, but a fixed hyperparameter (e.g., $\lambda=0.5$).
>
> Eq. (19) is an **implementation strategy**: instead of computing both losses every iteration, we sample one with probability $\lambda$, yielding an **unbiased estimator** of the same objective: L_BC(θ) = (1 − λ) L_comp + λ L_bnd ≈ E_{ξ ~ Bernoulli(λ)} [ (1 − ξ) L_comp + ξ L_bnd ].
>
> **W3.** Please refer to Table 5,6,7
>
> https://anonymous.4open.science/r/AnonymousICML2026-FFFFFF/Hyperparameters.pdf
>
> **W4.** We further increase the number of training seeds to **6** (each evaluated over 50 seeds), resulting in **300 total runs per result**. Please refer to the table for the new result.
> https://anonymous.4open.science/r/AnonymousICML2026-FFFFFF/MainResultAndAblation.pdf

---

> > ### Author Rebuttal · Reviewer_er13 · 2026-04-01
> >
> > I have increased my score.

---

### Decision · Program_Chairs · 2026-04-30

**Decision:**

Accept (regular)

**Comment:**

This paper introduces bootstrapped flow q-Learning (BFQL), a method designed to bypass the heavy computational costs of diffusion-based policies by enabling high-performance, single-step action generation in offline RL. While reviewers were initially concerned about the clarity of the mathematical notation and the method's relationship to existing consistency or shortcut models, the authors provided a robust rebuttal that included extensive new evaluations some benchmarks. They effectively demonstrated that their unique continuous-triplet consistency formulation and architectural preconditioning are essential for training stability and outperforming prior single-step baselines. Despite one reviewer’s lingering questions regarding the necessity of the velocity-field parameterization, the consensus shifted toward acceptance as the empirical results clearly show that BFQL offers a simpler, faster, and more expressive alternative to multi-step denoising. Overall, I recommend the paper for acceptance.